# The X-linked trichothiodystrophy-causing gene RNF113A links the spliceosome to cell survival upon DNA damage

Kateryna Shostak[1,2,11], Zheshen Jiang[1,2,11], Benoit Charloteaux[1,3,10,11], Alice Mayer[1,3], Yvette Habraken[1,4], Lars Tharun[5], Sebastian Klein[5], Xinyi Xu[1,2], Hong Quan Duong[1,2,6], Andrii Vislovukh[1,2], Pierre Close [1,7,8], Alexandra Florin[5], Florian Rambow[9], Jean-Christophe Marine[9], Reinhard Büttner[5] & Alain Chariot[1,2,8✉]

Prolonged cell survival occurs through the expression of specific protein isoforms generated by alternate splicing of mRNA precursors in cancer cells. How alternate splicing regulates tumor development and resistance to targeted therapies in cancer remain poorly understood. Here we show that RNF113A, whose loss-of-function causes the X-linked trichothiodystrophy, is overexpressed in lung cancer and protects from Cisplatin-dependent cell death. RNF113A is a RNA-binding protein which regulates the splicing of multiple candidates involved in cell survival. RNF113A deficiency triggers cell death upon DNA damage through multiple mechanisms, including apoptosis via the destabilization of the prosurvival protein MCL-1, ferroptosis due to enhanced SAT1 expression, and increased production of ROS due to altered Noxa1 expression. RNF113A deficiency circumvents the resistance to Cisplatin and to BCL-2 inhibitors through the destabilization of MCL-1, which thus defines spliceosome inhibitors as a therapeutic approach to treat tumors showing acquired resistance to specific drugs due to MCL-1 stabilization.

[1] Interdisciplinary Cluster for Applied Genoproteomics (GIGA), University of Liege, CHU, Sart-Tilman, Liège, Belgium. [2] Laboratory of Medical Chemistry, University of Liege, CHU, Sart-Tilman, Liège, Belgium. [3] GIGA Genomics Platform, University of Liege, CHU, Sart-Tilman, Liège, Belgium. [4] Laboratory of Gene Expression and cancer, University of Liege, CHU, Sart-Tilman, Liège, Belgium. [5] Institute for Pathology-University Hospital, Cologne, Germany. [6] Center for Molecular Biology, Institute of Research and Development, Duy Tan University, 03 Quang Trung, Danang, Vietnam. [7] Laboratory of Cancer Signaling, University of Liege, CHU, Sart-Tilman, Liège, Belgium. [8] Walloon Excellence in Life Sciences and Biotechnology (WELBIO), Wavres, Belgium. [9] Laboratory for Molecular Cancer Biology, VIB Center for Cancer Biology and KULeuven, Department of Oncology, 3000 Leuven, Belgium. [10]Present address: Department of Human Genetics, CHU of Liege, University of Liege, Sart-Tilman, 4000 Liège, Belgium. [11]These authors contributed equally: Kateryna Shostak, Zheshen Jiang, Benoit Charloteaux. ✉email: Alain.chariot@uliege.be

Spliceosomes are dynamic Ribonucleoprotein (RNP) complexes required for pre-mRNA splicing, i.e., the removal of non-coding intervening sequences (introns) from pre-mRNAs and for the ligation of coding sequences (exons) to generate mature mRNAs[1–3]. Intron excision occurs thanks to short sequence motifs in the pre-mRNA, at boundaries between the upstream exon and intron (the 5′ splice site) and the intron and downstream exon (the 3′splice site)[1,2]. RNA-protein, RNA-RNA and protein–protein interactions are critical for the proper recognition of splice sites. These interactions are dynamic during the splicing cycle and are critical for the formation, rearrangement and dissociations of the spliceosomal complexes[4].

Alternate splicing of mRNA precursors allows more than 95% of human genes to generate a variety of RNA species and distinct proteins from a single gene. Protein isoforms are selected by cancer cells to sustain survival and to promote tumor development and resistance to therapies[5]. Cancer cells exhibit global splicing deregulations due to mutations within mRNA sequences ("cis-acting mutations"), and in splicing factors ("trans-acting mutations") or to changes in expression levels of splicing factors[6,7]. As a result, the spliceosome has been defined as a promising therapeutic targets, as least for aggressive Myc-driven tumors and for malignant glioma[8,9]. Yet, transcripts whose splicing is deregulated in cancer cells only start to be identified[10].

The DNA damage response (DDR) helps the body to face thousands of DNA lesions[11,12]. These lesions, if not repaired, lead to mutations or genomic aberrations that threaten viability. DNA damage is induced by multiple sources ranging from byproducts of cell metabolism and oxidative damage to ionizing radiation and chemotherapeutic agents. Nucleotide excision repair (NER) repairs single-stranded DNA damage by removing helix-distorting DNA lesions induced by UV-light, ROS-induced cyclopurines and intrastrand crosslinks (ISCs) generated by chemotherapeutic drugs such as Cisplatin[13]. These ISCs activate several signal transduction pathways/kinases such as ATR[14]. ATR targets substrates including Chk1 in order to help tumor cells survive the DNA damage.

The double-strand breaks (DSBs) do not occur as frequently as DNA single-strand breaks (SSBs) but are more difficult to repair and therefore extremely toxic. The DDR includes the sensing of the broken DNA molecule, the activation of specific signaling pathways and finally, the repair of the DNA lesion[11]. DSB repair involves two main pathways, the homologous recombination (HR) and the non-homologous end-joining (NHEJ)[15,16]. In NHEJ, DSBs are sensed by the Ku70/80 heterodimer that recruits the DNA dependent protein kinase catalytic subunit (DNA-PKcs), Paralog of XRCC4 and XLF (PAXX) and end-processing enzymes leading to repair by the DNA ligase IV/DNA ligase IV-X-ray cross complementing protein 4 (XRCC4)/XRCC4-like factor (XLF, also referred to as Cernunnos) complex[11,17,18]. The recruitment of DNA-PKcs to DSBs occurs within seconds following DNA damage[19]. DSBs trigger DNA-PKcs autophosphorylation on the PQR cluster that includes Ser2056 and on the ABCDE cluster of 6 amino acid residues between Thr2609 and Thr2647, which is also targeted by the kinase ATM upon DNA damage[20–22]. These phosphorylations alter the affinity of DNA-PKcs for DNA and inactivate its kinase activity to promote NHEJ[23–25].

Cancer cells show genomic instability[26,27]. Although the DDR is commonly activated in early neoplastic lesions as a protective mechanism against malignancy, cancer cells overcome this barrier through the mutational or epigenetic inactivation of DDR components to enhance cell proliferation and survival, despite increased genomic instability[28–31].

We show here that RNF113A, also referred to as Cwc24 in yeast, whose deficiency causes a novel X-linked trichothiodystrophy (TTD) and enhances sensitivity to Interstrand DNA Crosslinking agents in C.elegans[32,33], is aberrantly expressed in pulmonary adenocarcinomas. RNF113A promotes cell survival in Cisplatin-treated lung cancer-derived cells as a subunit of the spliceosome. We characterized all transcripts whose splicing relies on RNF113A and also established a link between RNF113A and MCL-1 stabilization, with important consequences for the treatment of lung cancer cells showing some acquired resistance to BCL-2 inhibitors. Therefore, our data define RNF113A as a promising therapeutic target.

## Results

**RNF113A is increased in pulmonary adenocarcinomas.** In a search for E3 ligases overexpressed in cancer, we got interested in RNF113A which is detected in all human cases of haematological or solid tumors (http://www.proteinatlas.org/ENSG00000125352 –RNF113A/cancer). RNF113A expression was higher in our clinical cases of pulmonary adenocarcinomas than in tumor-free lung parenchymas, independently of the K-RAS or EGFR mutational status (Fig. 1a and Supplementary Data 1). A majority of lung malignancies showed a mostly nuclear staining of RNF113A (Fig. 1b). A weak and almost exclusively nuclear staining of RNF113A was detected in normal lung epithelium cells (Supplementary Fig. 1a). Importantly, patients with high levels of RNF113A showed a shorter survival rate (Fig. 1c). RNF113A expression increased at both mRNA and protein levels in Cisplatin-treated A549 and BZR-T33 adenocarcinoma-derived cells (Fig. 1d, e). Consistently, RNF113A expression was barely detectable in untreated A549 cells but was detected in Cisplatin-treated cells by immunofluorescence analyses (Fig. 1f). Camptothecin, a topoisomerase I inhibitor, also increased RNF113A expression in both A549 and BZR-T33 cells while Etoposide, a topoisomerase II inhibitor, did not (Fig. 1g, h and Supplementary Fig. 1b, respectively). RNF113A expression was also weakly induced by γ-irradiation (Supplementary Fig. 1c). RNF113A expression was also induced by Cisplatin in normal human dermal fibroblasts (Supplementary Fig. 1d). Therefore, RNF113A expression is induced by some DNA-damaging signals and is increased in lung cancer.

CCAAT/enhancer-binding protein beta (C/EBPβ) expression is induced by DNA-damaging agents[34]. Therefore, we investigated whether C/EBPβ is required for the induction of RNF113A expression by Cisplatin. C/EBPβ deficiency interfered with Cisplatin-dependent induction of RNF113A in A549 cells (Fig. 1i). Consistently, we found C/EBPβ binding sites on the RNF113A promoter using the TFbind software (http://tfbind.hgc.jp/) (Fig. 1j). C/EBPβ was recruited on site 1 in unstimulated A549 cells and on sites 1 to 4 in Cisplatin-treated cells (Fig. 1j). p53 was dispensable for RNF113A expression as the incubation of A549 cells with Nutlin, which disrupts the interaction of the E3 ligase MDM2 with p53, or with JNJ26854165, a MDM2 inhibitor[35], did not impact on RNF113A expression (Fig. 1k). Therefore, Cisplatin induces the expression of RNF113A through a C/EBPβ-dependent but p53-independent pathway.

**RNF113A protects from Cisplatin-dependent cell death.** We next explored whether RNF113A is involved in the DDR. Enhanced RNF113A expression in A549 cells interfered with Cisplatin-dependent DNA-PKcs phosphorylation on Ser2056, a marker of DNA damage (Fig. 2a). RNF113A overexpression protected A549 cells from Cisplatin-induced death (Fig. 2b). On the other hand, RNF113A deficiency enhanced cell death in Cisplatin-treated lung cancer A549 and BZR-T33 cells (Fig. 2c and Supplementary Fig. 2a). RNF113A deficiency did not impact on p53 phosphorylation in BZR-T33 cells triggered by Cisplatin

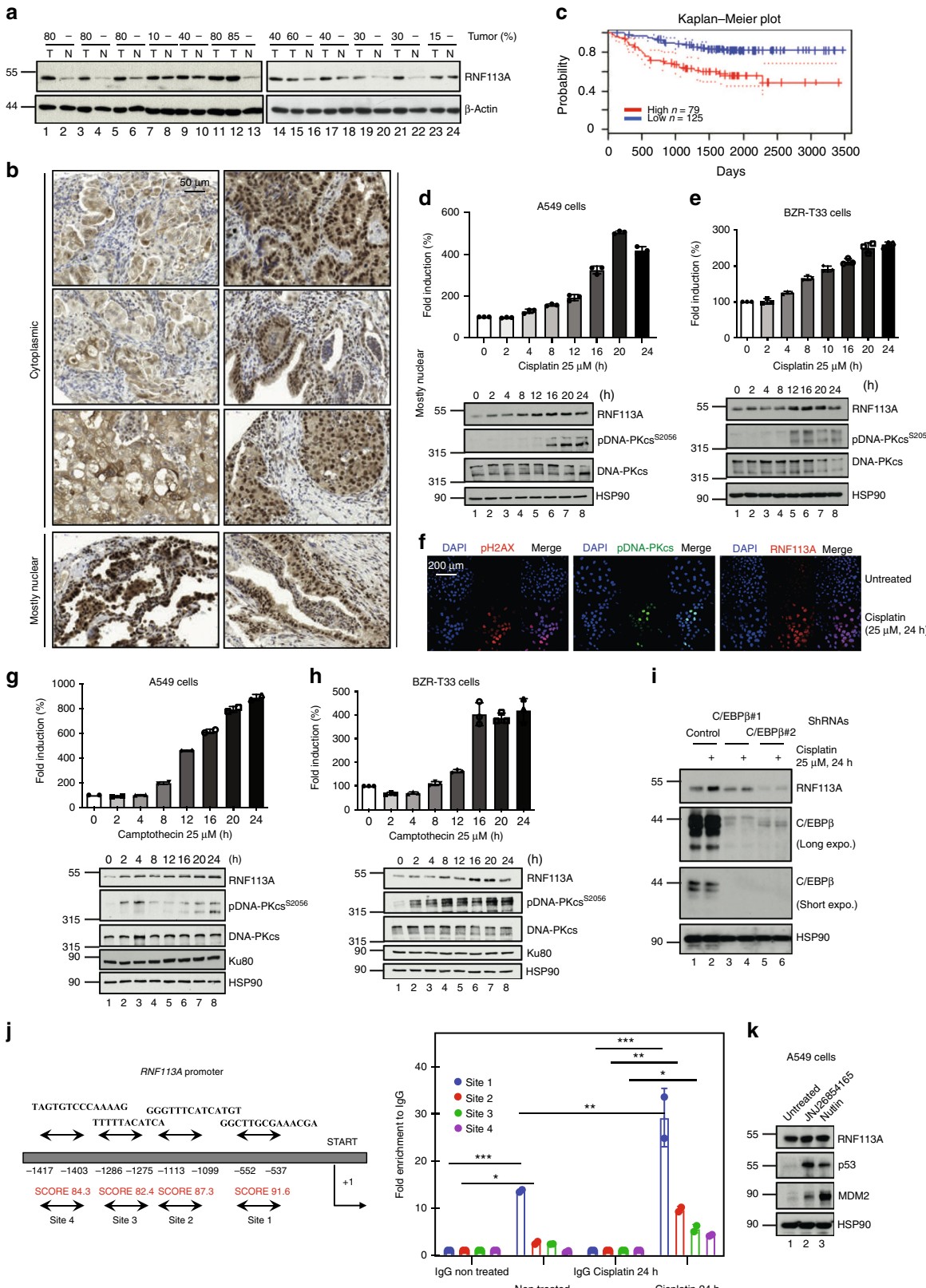

(Fig. 2d). Cisplatin-dependent DNA-PKcs phosphorylation on S2056 was increased upon RNF113A deficiency in BZR-T33, A549 and HT1975 cells showing distinct p53 status (Fig. 2d, Supplementary Fig. 2b and Supplementary Fig. 2c). Accordingly, RNF113A deficiency enhanced the number of both phospho-$H_2AX$ (pH2AX) and phospho-DNA-PKcs (pDNA-PKcs) positive

BZR-T33 cells, suggesting that these cells fail to repair DNA (Fig. 2e, f). RNF113 overexpression also protected A549 cells from cell death induced by Etoposide and limited DNA-PKcs phosphorylation on serine S2056 (Supplementary Fig. 3a). Consistently, cell death triggered by Etoposide was more pronounced upon RNF113A deficiency in A549 cells (Supplementary Fig. 3b).

**Fig. 1 RNF113A expression in lung cancer and upon Cisplatin treatment. a** RNF113A expression in lung cancer. Western blot (WB) analyses with human lung tumors ("T") and corresponding normal adjacent tissues ("N"). Numbers refer to the percentage of tumor cells. **b**. Subcellular localizations of RNF113A in lung adenocarcinomas. Anti-RNF113A Immunohistochemistry analyses were conducted on a Tissue-MicroArray (TMA) (×40 objective). Stromal cells served as internal negative control. **c** Patients with high levels of RNF113A mRNAs show a shorter survival rate (Kaplan-Meir plot). **d, e** RNF113A expression is induced by Cisplatin in A549 (**d**) or BZR-T33 (**e**) cells. On the top, RNF113A mRNA levels in unstimulated cells is set to 100% and levels in other experimental conditions are relative to that after normalization with β-actin. Data from two independent experiments performed in triplicates (means ± SD) are shown. At the bottom, WB analyses. The anti-pDNA-PKcs (S2056) antibody was used to prove DNA damage. **f** Induction of RNF113A expression by Cisplatin in A549 cells (immunofluorescence analyses). **g, h** RNF113A expression is induced by Camptothecin in A549 (**g**) or BZR-T33 (**h**) cells. On the top, data from two Real-time PCR (independent) analyses performed in triplicates (means ± SD) are plotted as described in (**d**). At the bottom, WB analyses. **i** RNF113A induction by Cisplatin occurs through a C/EBPβ-dependent pathway. Control or C/EBPβ-depleted A549 cells were treated or not with Cisplatin (25 μM) for 24 h and WB analyses were done. Expo. = exposure. **j** C/EBPβ is recruited on the *RNF113A* promoter. C/EBPβ binding sites were identified (Tfbind software) and ChIP assays using an anti-C/EBPβ antibody were carried out. Histogram show recruitment C/EBPβ on indicated sites with or without treatment (IgG antibody was used as negative control). RNF113A promoter is lacking a TATA box. Results of two independent experiments (means ± SD, Student t-test, ***p < 0.001, **p < 0.01, *p < 0.05) are shown. START = start of transcription. **k** RNF113A expression is not regulated by p53. A549 cells were incubated or not with JNJ-2685416 or with Nutlin and WB analyses were done. Source data for all figures are provided as a Source Data file.

If cells are allowed to resume proliferation after being stimulated with Cisplatin for 16 h, ATR activation assessed through phosphorylation of its target Chk1, was also defective upon RNF113A deficiency in A549 cells (Fig. 2g). RNF113A-depleted cells underwent Caspase 3-dependent cell death upon DNA damage (Fig. 2g). The ability of control versus RNF113A-deficient BZR-T33 cells to undergo DNA repair was assessed with the comet assay. RNF113A-deficient cells showed more DNA damage, especially after Cisplatin treatment, as assessed through the quantification of the tail moment (Fig. 2h). Thus, RNF113A promotes DNA repair.

**RNF113A is recruited on some DNA damage-induced foci**. We next explored whether RNF113A is recruited on DNA damage foci, using a RNAse A-based extraction protocol to visualize the formation of Cisplatin-induced foci on damaged DNA. RNF113A colocalized with phosphorylated DNA-PKcs but not with phospho-H2AX foci in Cisplatin-treated cells (Supplementary Figs. 4a, b, respectively). Consistently, RNF113A was found in chromatin fractions in BZR-T33 cells treated to Cisplatin (Fig. 3a). RNF113A-deficient cells also had more Ku70/80 and DNA-PKcs recruited to chromatin fractions upon Cisplatin treatment (Fig. 3b). Conversely, RNF113A-overexpressing A549 cells had less Ku70/80, DNA-PKcs and DNA ligase IV in chromatin fractions when treated with Cisplatin (Fig. 3c and Supplementary Fig. 4c), suggesting that RNF113A controls the recruitment of DNA repair factors on chromatin. To explore whether RNF113A regulates the presence of phospho-H2AX on DSB sites, we used the DIvA U2-OS cell line, which stably expresses the restrictase AsiSI under *ER* promoter. These cells generate several randomly distributed and sequence-specific DSBs[36]. Treatment of this cell line with 4-hydroxy tamoxifen (4OHT) generated DSBs since multiple pH2AX[+] cells were detected by immunofluorescence (Supplementary Fig. 5). We therefore generated control and RNF113A-depleted cells (Supplementary Fig. 5). ChIP assays were conducted to assess the presence of pH2AX on AsiSI sites in both control and RNF113A-depleted cells using appropriate primers[36]. pH2AX on H2AX-associated AsiSI sites using primers 183, 906, 307 and 221[36] was defective upon RNF113A deficiency (Fig. 3d). As negative controls, we also conducted these experiments using primers 811 and 903, which are not H2AX-associated AsiSI sites (Fig. 3d)[36]. Therefore, RNF113A controls the pool of NHEJ factors recruited to damaged DNA.

In support with a nuclear localization of RNF113A, wild type RNF113A and a mutant lacking the first 30 N-terminal amino acids ("ΔN30") or the RING domain ("ΔRING", which lacks amino acids from 262 to 300) were mostly found in the nucleus

and colocalized with DNA-PKcs when transfected in A549 cells (Fig. 3e). On the other hand, a RNF113A construct lacking the first 60 or 90 N-terminal amino acids ("ΔN60 and ΔN90") mostly showed both nuclear and cytoplasmic localizations (Fig. 3e). A predicted nuclear localization signal (NLS) was found on RNF113A, using the SeqNLS algorithm[37] (Fig. 3e). Despite the fact that RNF113A was found in the nucleus of lung cancer cells, Cisplatin enriched the cytoplasmic pool of RNF113A (Fig. 3a). This was also true in BEAS-2B cells, which are derived from a normal bronchial epithelium (Supplementary Fig. 4d). Consistently, cells showing DNA damage (i.e., positive for pH2AX) and undergoing apoptosis (i.e., in which Cyt. C was released) showed a cytoplasmic staining of RNF113A (Fig. 3f). Therefore, DNA damage and cell apoptosis triggered by Cisplatin correlates with the shuttling of RNF113A to the cytoplasm.

**RNF113A acts as a subunit of the spliceosome**. As RNF113A is a spliceosome subunit[4,38], we carried out RNA immunoprecipitation (RIP) experiments in A549 cells and looked for spliceosome subunits. We first fixed cells with paraformaldehyde (PFA) to trap large protein complexes onto RNAs. RNF113A, SF3B1 (another spliceosome subunit) and the ribosomal protein RPL7 but not HSP90 were found on RNAs (Fig. 4a). Next, we cross-linked our cells with UV to discriminate direct versus indirect bindings of spliceosome subunits to RNAs. In those circumstances, RNF113A bound mRNAs in unstimulated cells which were not treated with RNAse and this binding was negatively regulated by Cisplatin, which fits with our previous data showing that a pool of RNF113A moves into the cytoplasm upon DNA damage (Fig. 4b). As expected, SF3B1 as well RPL7 were also found associated to RNAs and SF3B1 binding to RNAs was also impaired upon Cisplatin stimulation (Fig. 4a, b). Therefore, RNF113A directly binds RNAs.

Because a displacement of spliceosomes causes the accumulation of three-strand nucleic acid structures formed by an RNA: DNA hybrid plus a displaced DNA strand (ssDNA) and referred to as "R-loops"[39], we next explored whether RNF113A deficiency leads to the accumulation of R-loops. RNF113A-depleted A549 cells indeed showed more R-loops, as judged by immunofluorescence analyses using an antibody that specifically detects these R-loops (Fig. 4c). Protein levels of the spliceosome factor SF3B2 were decreased upon RNF113A deficiency in A549 cells (Fig. 4d)[4]. Moreover, protein levels of DNA repair factors such as RNF8 and Rad51, which are downregulated upon impaired splicing[40], were also decreased in RNF113A-depleted A549 cells (Fig. 4d). Although SF3B2 levels did not decrease upon RNF113A deficiency in BZR-T33 cells, in contrast to A549 cells, SF3B2

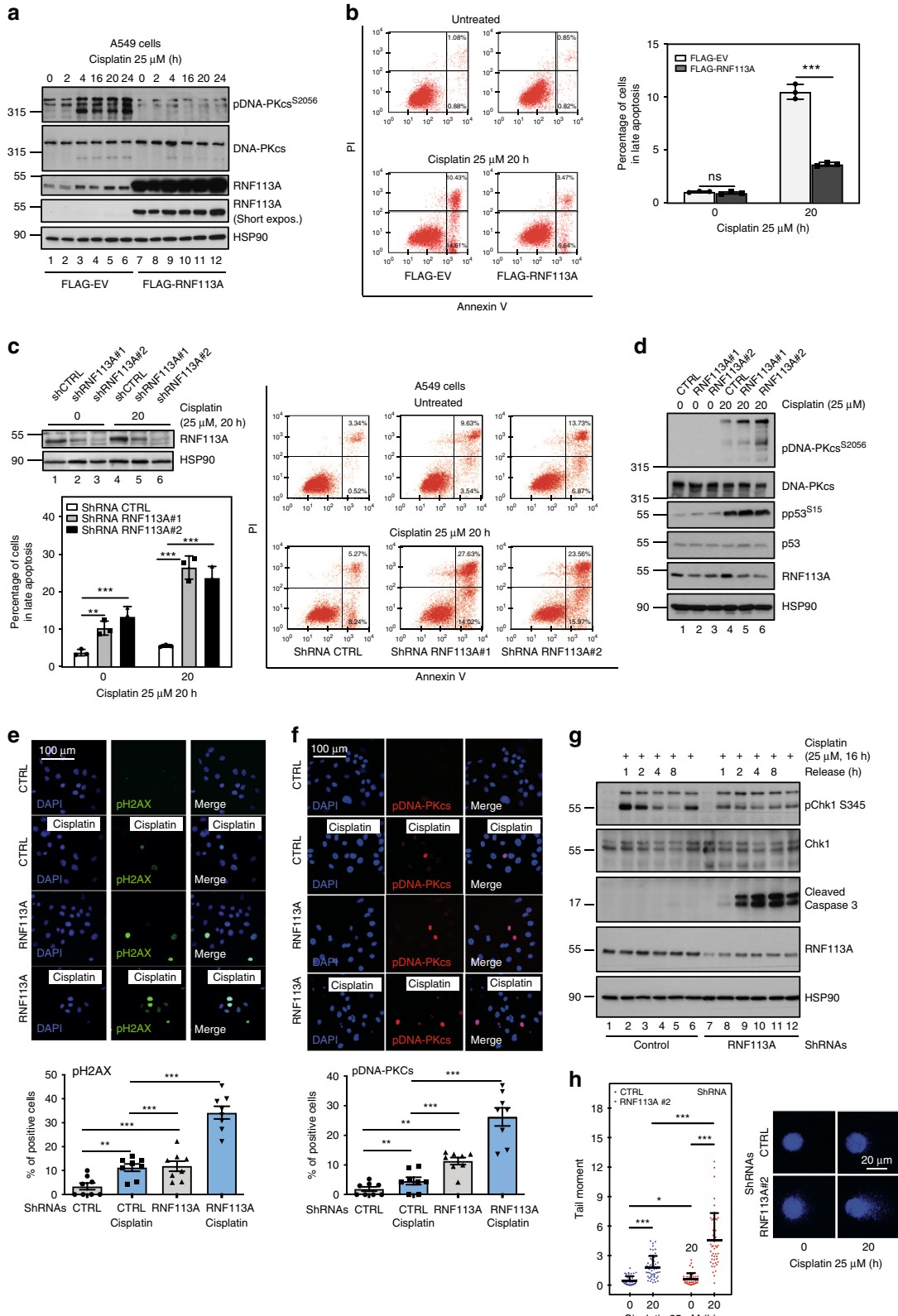

binding to SF3B1 was nevertheless impaired in both cell lines, suggesting that the assembly of spliceosome subunits relies on RNF113A (Fig. 4e, f). Consistently, R-loops accumulated in SF3B2-depleted A549 cells (Fig. 4g). Moreover, SF3B2 deficiency also led to more DNA damage, as evidenced by enhanced phosphorylated levels of DNA-PKcs and to enhanced cell death in A549 cells treated or not with Cisplatin (Fig. 4h, i, respectively). Finally, both SF3B1 and SF3B2 were found in the cytoplasm and in the nucleus of A549 cells but Cisplatin promoted their disengagement from chromatin, as previously seen for RNF113A (Fig. 4j). Therefore, the spliceosome subunit SF3B2 promotes cell survival upon DNA damage, similarly to RNF113A.

**Fig. 2 RNF113A limits Cisplatin-dependent cell death. a** RNF113A overexpression interferes with DNA-PKcs phosphorylation upon Cisplatin treatment. Control or RNF113A-overexpressing A549 cells were stimulated or not with Cisplatin and WB analyses were done. **b** RNF113A overexpression limits Cisplatin-dependent cell death. Control or RNF113A-overexpressing A549 cells were untreated or stimulated with Cisplatin. The percentage of cells in early (Annexin V positive and PI negative) or late apoptosis (Annexin V positive and PI positive) was assessed by FACS. On the left, FACS data from one representative experiment. On the right, the histogram from two independent experiments (Student t-test, p-values: ***<0.001). ns = non significant. **c** RNF113A deficiency enhances Cisplatin-mediated cell death. Extracts from control ("ShCTRL") or from RNF113A-depleted ("ShRNF113A#1 and ShRNF113A#2") A549 cells were subjected to WB analyses. Cell survival upon Cisplatin treatment was assessed by FACS (right panel). On the left, FACS data from two independent experiments are illustrated in the histogram (Student t-test, ***p < 0.001). **d** RNF113A negatively regulates Cisplatin-induced DNA-PKcs phosphorylation. Control or RNF113A-depleted BZR-T33 cells were untreated or stimulated with Cisplatin and WB analyses were done. **e**, **f**. RNF113A deficiency enhances the number of $pH_2AX$ (S139) (**e**) and pDNA-PKcs (S2056)$^+$ (**f**) cells upon Cisplatin treatment. Control and RNF113A-deficient A549 cells were treated with Cisplatin (25 μM) for 4 h and immunofluorescence analyses were done to quantify the number of $pH_2AX^+$ or pDNA-PKcs (S2056)$^+$ cells. The corresponding histogram represents 10 blindly taken fields containing at least 150 nuclei per field (Student t-test, ***p < 0.001, **p < 0.01). **g** RNF113A deficiency impairs ATR activation upon DNA damage. Control or RNF113A-depleted A549 cells were treated or not with Cisplatin and then allowed to grow in a fresh media for the indicated periods of time. Chk1 phosphorylation was assessed by WB. **h** RNF113A promotes DNA repair. Control versus RNF113A-depleted BZR-T33 cells were treated with Cisplatin and an alkaline Comet Assay was done. Fifty images per conditions were analyzed by OpenComet in ImageJ. The tail moment of every cell was calculated in control versus RNF113A-depleted BZR-T33 cells. Data are shown as mean ± SD (***p < 0.001, *p < 0.05, Student t-test).

To investigate the impact of RNF113A depletion on both gene expression and splicing events, we first concentrated on candidates such as Rad51 and RNF8 which are regulated by splicing[40]. RNF113A-depleted cells accumulated Rad51 and RNF8 pre-mRNAs in which intron 1 is retained, especially after Cisplatin stimulation (Supplementary Figs. 6a and b, respectively). Therefore, RNF113A is required for the splicing of both Rad51 and RNF8. SF3B2-depleted cells also accumulated Rad51 pre-mRNAs, especially after Cisplatin treatment (Supplementary Fig. 6a). As RNF113A-depleted cells undergo cell death upon Cisplatin stimulation, we next wondered whether Caspase 3 activation is the causal event rather than the consequence of the splicing deregulations seen upon RNF113A deficiency. To address this issue, we pretreated A549 cells with ZVAD, a caspase inhibitor, and assessed Rad51 and RNF8 splicing upon Cisplatin stimulation in both control and RNF113A-depleted cells. ZVAD did not impact on the accumulation of both Rad51 and RNF8 pre-mRNAs upon RNF113A deficiency in Cisplatin-treated cells (Supplementary Fig. 6a, b, respectively). Therefore, apoptosis is the consequence rather than the cause of the splicing deregulations seen upon RNF113A deficiency in cells treated with Cisplatin.

We expanded this analysis genomewide by carrying out high-throughput RNA sequencing on total RNA extracted from control and RNF113A-depleted A549 cells treated or not with Cisplatin. Having performed two sequencing runs on the pooled mRNA libraries to obtain a minimum of 45 millions of paired-end reads per sample, we had sufficient coverage of lowly abundant transcripts to perform analyses of the resulting dataset at several levels: (i) a differential gene expression analysis to evaluate the expression changes induced by the different treatments (RNF113A depletion and/or Cisplatin) at the gene level; (ii) a differential gene splicing analysis assessing individual splicing events categorized by type (i.e., skipped exon, retained intron, alternative 5′ splicing site, alternative 3′ splicing site, and mutually exclusive exons); and (iii) a global analysis of intron retention at the gene level.

Differential expression analysis showed that the expression level of many genes is affected by Cisplatin treatment and/or by RNF113A depletion (Supplementary Fig. 7a, b, and Supplementary Data 2). For the majority of these candidates, expression changes can mostly be explained by an additive model where Cisplatin treatment and RNF113A depletion have independent effects, with 3703 genes whose mRNA expression levels were significantly changed upon Cisplatin treatment (1722 upregulated and 1981 downregulated; FDR q-value < 5% and twofold change

minimum) and 1816 genes significantly affected by RNF113A depletion (866 upregulated and 950 downregulated; FDR q-value < 5% and twofold change). For a few genes (293), the interplay of the two treatments is more complex with a contrasted impact of the depletion between Cisplatin-treated and untreated cells. This was the case for instance for NUPR1 for which RNF113A depletion drastically reduced the increase in mRNA expression level induced by Cisplatin. Although NUPR1 mRNA expression level increased by 5.6 fold in control cells treated with Cisplatin, this increase was only of 1.5 fold in RNF113A-depleted cells (Supplementary Data 2).

To verify the validity of our RNAseq dataset, we performed a gene set enrichment analysis (GSEA). As expected, this analysis highlighted a significant enrichment in signatures typical of apoptosis and DNA repair (FDR q-value < 1%) (Supplementary Fig. 7c). We also confirmed in independent Real-Time PCR experiments the expression changes observed for a few candidates highlighted in the systematic differential expression analysis. TRIM29 and CEACAM5 expression were increased upon RNF113A deficiency while GFRA1 expression was decreased (Supplementary Fig. 7d).

We then focused on the role of RNF113A in splicing using rMATS (replicate Multivariate Analysis of Transcript Splicing), a computational method dedicated to the detection of differential splicing events from replicate RNAseq data[41]. Using a hierarchical model to account for uncertainty in each replicate and variability between replicates, rMATS assesses individual splicing events from five different categories: skipped exons, retained intron, alternative 5′ splicing site, alternative 3′ splicing site, and mutually exclusive exons (Fig. 5a).

To evaluate the impact of Cisplatin and/or RNF113A depletion on splicing, we performed four different comparisons, each evaluating the impact of one factor, the other one kept constant (Fig. 5a, b). The most frequent categories of splicing events evaluated by rMATS involved exons (Skipped exons and mutually exclusive exons) but several thousands of events were also evaluated for each one of the other categories (Fig. 5b, Supplementary Fig. 8a and Supplementary Data 3-8). Significant alternative splicing events (FDR q-value < 5% and difference in inclusion level > 20%) were detected for all comparisons and for all categories of events. Cisplatin treatment in RNF113A-depleted cells was associated with the highest number of significant exon skipping events (1342 exons show significantly higher inclusion levels when depleted cells are treated with Cisplatin). RNF113A depletion was shown to induce smaller changes on exon retention, although hundreds of events were significantly affected.

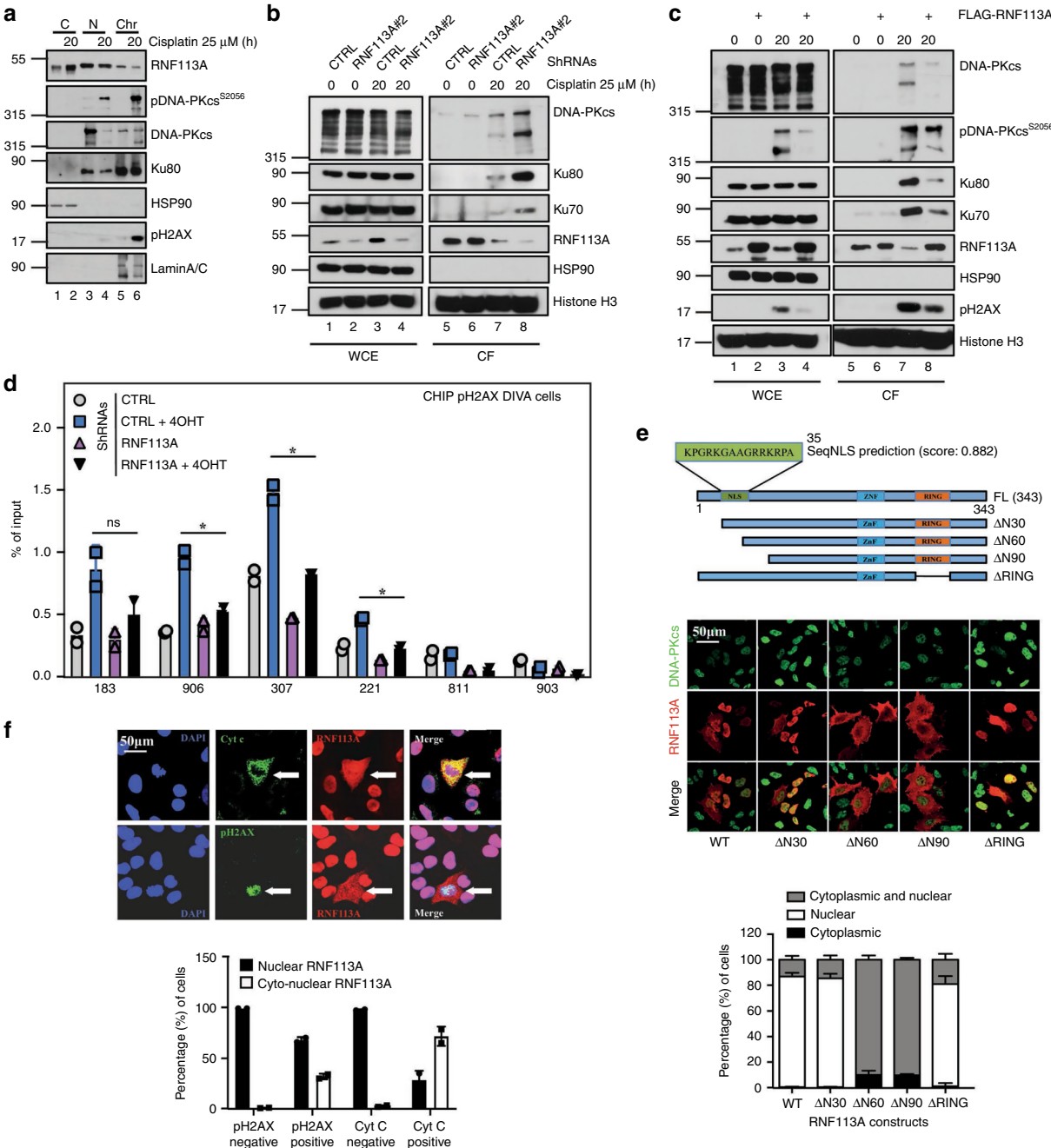

**Fig. 3 RNF113A is recruited on DNA damage-induced foci. a** RNF113A is in both the cytoplasm and the nucleus. A549 cells were treated or not with Cisplatin and WB analyses were carried out with cytoplasmic, nuclear and chromatin-enriched extracts. **b**, **c** RNF113A controls the recrutment of NHEJ factors on chromatin upon DNA damage. Control versus RNF113A-depleted A549 cells (**b**) or control versus RNF113A-overexpressing A549 cells (**c**) were treated or not with Cisplatin and WB analyses were carried out on chromatin fractions after pre-extraction with the CSK + RNase A buffer. **d** RNF113A controls the recruitment of $pH_2AX$ on DSBs. ChIP assays were conducted with extracts from control and RNF113A-depleted DIvA U2-OS cells treated or not with 4-hydroxy Tamoxifen (TAM). Primers 183, 906, 307, and 221 are $pH_2AX$-associated AsiSI sites while primers 811 and 903 are non-associated and serve as negative control[36]. Immunoprecipitations using anti-IgG antibody served as negative control. The histogram shows recruitment of $pH_2AX$ on indicated sites. Results of two independent experiments (means ± SD, Student *t*-test, *$p < 0.05$) are shown. **e** A N-terminal nuclear localization signal (NLS) controls the nuclear import of RNF113A. The human RNF113A sequence was analyzed using the online NLS prediction algorithm SeqNLS (http://mleg.cse. sc.edu/seqNLS/) and the identified NLS (residues 21–35) is shown in green with a possibility score of 0.882. The subcellular localization of RNF113A constructs was analyzed by immunofluorescence in A549 cells, using DNA-PKcs as a nuclear marker. The percentage of cells showing a cytoplasmic and/ or nuclear localization of RNF113A constructs is illustrated in the histogram below. **f** RNF113A moves in the cytoplasm of apoptotic cells showing some DNA damage. Anti-RNF113A Immunofluorescence analyses were conducted in Cisplatin-treated A549 cells. Cells showing some DNA damage or undergoing cell apoptosis were identifed through anti-$pH_2AX$ and Cyt C stainings, respectively. The histogram show the percentage of cells showing a nuclear or a cytoplasmic and nuclear localization of RNF113A, depending on their status for pH2AX or for Cyt C.

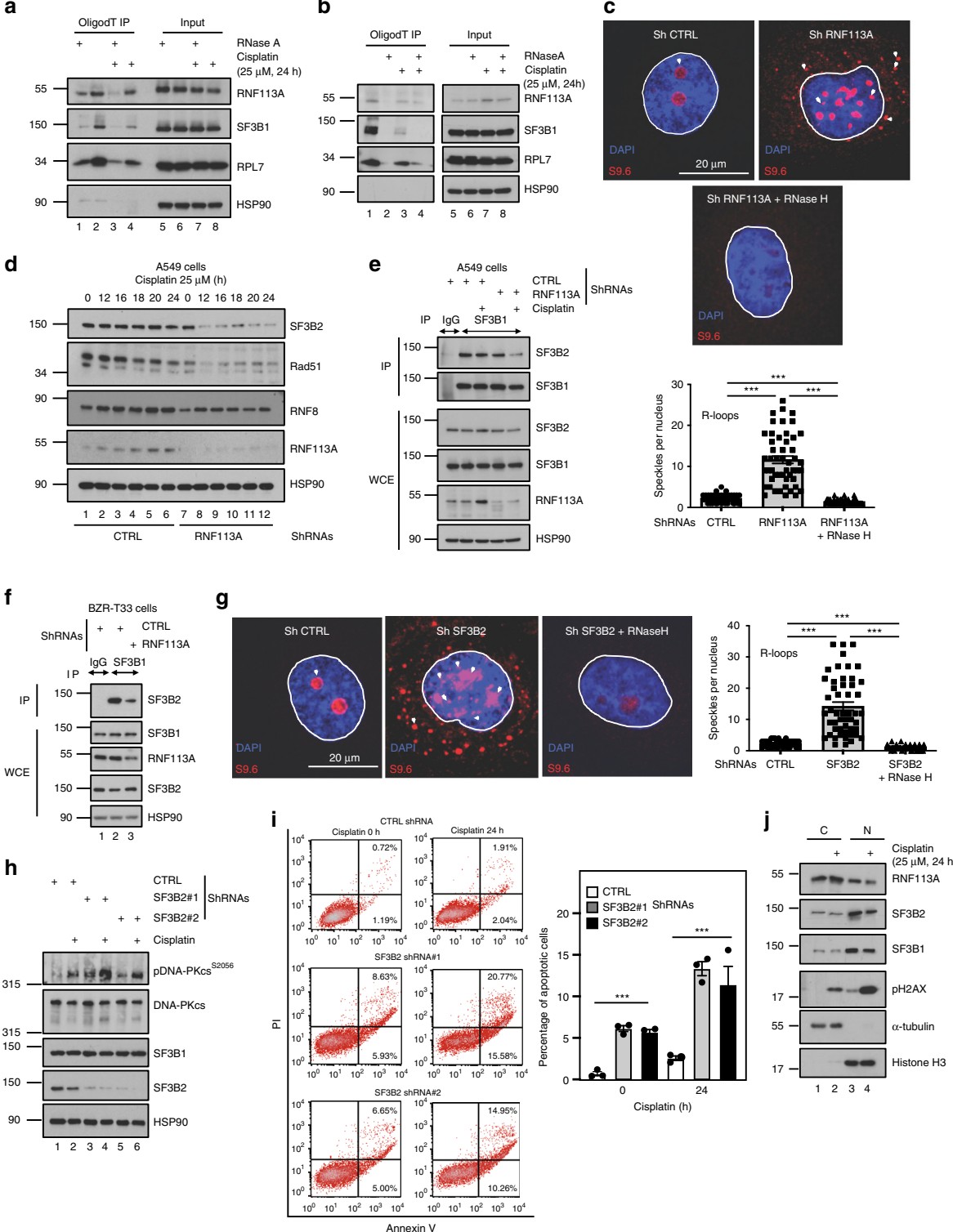

Depletion of RNF113A in Cisplatin-treated cells was associated with much higher intron retention levels (1397 introns with significantly higher inclusion levels in depleted samples).

When normalizing for the number of events evaluated in each category, intron retention was by far the most frequently affected category (Fig. 5c). The most frequent differences were observed upon RNF113A depletion, with 22% and 34% of the events showing a significant intron retention increase in untreated and in Cisplatin-treated cells, respectively. Although cisplatin induced only a few changes in intron retention level in control cells, the same treatment induced more intron retention in RNF113A-depleted cells, consistent with a role of RNF113A in intron splicing. Almost all events significant upon depletion were associated with an intron retention level higher in depleted than in non-depleted cells (Fig. 5d, and Supplementary Fig. 8a). In contrast, the impact of Cisplatin on intron retention was more balanced with both higher and lower intron inclusion levels depending on the event considered, especially in control cells (Fig. 5d and Supplementary Fig. 8a).

**Fig. 4 RNF113 is a RNA-binding protein. a, b** RNF113A is recruited on RNAs. Unstimulated or Cisplatin-treated A549 cells were treated with paraformaldehyde to fix larger complexes (**a**) or irradiated with UV to covalently crosslink direct RNA-protein interaction (**b**), incubated or not with RNase H and RNA immunoprecipitations using OligodT magnetic beads were done followed by WB analyses (left lanes). Cell extracts before RNA Immunoprecipitations were also subjected to WB analyses ("Input"). **c** RNF113A-depleted lung cancer cells accumulate R-loops. Control or RNF113A-depleted A549 cells were subjected to immunofluorescence analyses using the anti-R-loops antibody (63X objective lens). A quantification of DNA-RNA hybrids is illustrated. For quantification, nuclei specles in a total of 50 nucleus in each experimental condition were counted (Student t-test, ***p < 0.001). **d** RNF113A promotes SF3B2, Rad51 and RNF8 expression. Control or RNF113A-depleted A549 cells were untreated or stimulated with Cisplatin and WB analyses were done. **e, f** RNF113A is required for the integrity of the spliceosome. Control or RNF113A-depleted A549 (**e**) or BZR-T33 (**f**) cells treated or not with Cisplatin were subjected to anti-IgG (negative control) or -SF3B1 immunoprecipitations followed by anti-SF3B2 WBs. Cell extracts were also subjected to WBs (lower panels). **g** SF3B2-depleted lung cancer cells accumulate DNA-RNA hybrids. Control or SF3B2-depleted A549 cells were subjected to immunofluorescence analyses as described in (**c**). **h** SF3B2 deficiency enhances DNA-PKcs phosphorylation upon Cisplatin treatment. Control and SF3B2-depleted A549 cells were treated or not with Cisplatin (25 µM for 24 h) and WB analyses were done. **i** SF3B2 deficiency enhances cell death upon DNA damage. On the left, cell survival upon Cisplatin treatment in control and SF3B2-depleted A549 cells was assessed by FACS. The percentage of cells in early or late apoptosis is quantified. On the right, FACS data from two independent experiments are illustrated (Student t-test, ***p < 0.001). **j** SF3B1, SF3B2, and RNF113A moves from the nucleus to the cytoplasm of lung cancer cells upon DNA damage. A549 cells were treated or not with Cisplatin and WB analyses were done (cytoplasmic and nuclear extracts).

Athough the analysis with rMATS clearly shows an impact of RNF113A depletion on all types of splicing events and most notably on intron retention, there are two potential caveats in this analysis. First, rMATS evaluate various number of events per gene. Second, only a small fraction of intron retention events is analyzed as introns are generally not described in genome annotations. To verify that the above results are not affected by these caveats, we designed an alternative strategy measuring specifically the coverage of intronic versus exonic regions for all transcribed protein-coding genes detected in our transcriptomic profiles. Strikingly, RNF113A depletion induced an increase of the relative fraction of reads mapping to intronic regions for most genes and the effect was even more pronounced in Cisplatin-treated cells (Fig. 5e, f). This increase was significant when comparing the distributions of all genes (Supplementary Fig. 8b) or global measures for the entire transcriptome (Fig. 5f). Although this effect was less obvious for genes with lower expression levels (Supplementary Fig. 8c), the impact of RNF113A depletion appeared to be largely unspecific and to affect most genes. Therefore, beyond an expected impact on the gene expression, the splicing is globally affected. Although some specific exon skipping events are significant upon RNF113A silencing, the main impact of the knockdown is a slight but significant increase of intron retention. These observations are consistent with the reported role of RNF113A as a spliceosome subunit.

**RNF113A controls the splicing of pro-survival candidates.** We next concentrated on differential splicing of candidates found in our RNA-Seq analyses. *SAT1* was a candidate impacted by the treatments both at the transcriptional and at the splicing level. Indeed, *SAT1* expression level significantly increased upon Cisplatin treatment (Log2 fold change = 1.5; adjusted *p*-value = 1.1E −23; Supplementary Table 2) while its splicing was affected specifically upon RNF113A deficiency in Cisplatin-treated cells. *SAT1* gene has 7 exons. Exon 4 contains a premature STOP codon and need to be skipped to give rise to a mRNA coding for a functional protein (Fig. 6a). The analysis performed with rMATS showed that in control cells treated or not with Cisplatin and in untreated RNF113A-depleted cells, exon 4 is included in roughly half of the transcripts (Fig. 6b, left panel). In contrast, the inclusion level of exon 4 was significantly lower in RNF113A-depleted cells treated with Cisplatin (Fig. 6b, left panel). Combined with the effect of Cisplatin on *SAT1* expression, the modifications in exon 4 inclusion level leads to major differences in abundance of the two corresponding transcripts between the four sets of conditions (Fig. 6b, middle and right panels). Cisplatin

increased the expression of the SAT1-encoding mRNA in both control and RNF113A-depleted cells (Fig. 6b, right panel). While Cisplatin also induced an increase in exon 4-containing mRNAs in control cells, this increase was however abolished in RNF113A-depleted cells (Fig. 6b, middle panel). Quantification of *SAT1* mRNAs by RT-PCR confirmed the increase of the SAT1-coding transcript in the RNF113A-depleted cells treated with Cisplatin (Fig. 6c, lower panels). Apoptosis was not the causal event of this process as RNF113A-depleted cells treated with Cisplatin in which Caspase 3 activation was prevented by ZVAD still showed elevated levels of the SAT1-coding transcript (Fig. 6c). SF3B2 deficiency also caused an increase of the SAT1-coding transcript upon treatment with Cisplatin, demonstrating again that SF3B2 and RNF113A deficiencies share many similarities (Fig. 6d). Moreover, ZVAD did not prevent these splicing deregulations seen upon Cisplatin treatment of SF3B2-depleted cells (Fig. 6d). As a result of this increase of the SAT1-coding transcript, protein levels of SAT1 dramatically increased in RNF113A-depleted A549 cells treated with Cisplatin (Fig. 6e). As SAT1 promotes ferroptosis[42], we reasoned that RNF113A-depleted cells may undergo ferroptosis upon stimulation with Cisplatin. RNF113A deficiency indeed potentiated the production of lipid ROS, which accumulate in cells undergoing ferroptosis, upon Cisplatin stimulation, as evidenced by the quantification of BODIPY-C11 staining (Fig. 6f).

If *NUPR1* expression is increased by Cisplatin in control cells, this induction is severely defective in RNF113A-deficient cells (Fig. 7a, b). This effect on *NUPR1* mRNA expression level was observed both in our high-throughput RNA sequencing data and by real-time PCR (left and right panels, respectively). To check for a specific impact on *NUPR1* splicing, we designed a RT-PCR experiment with primers targeting regions in exons 1 and 2. We detected two *NUPR1* transcripts, a first transcript properly spliced (all intron removed) and translated into the NUPR1 protein, as well as a second transcript in which intron 1 was not removed, which presumably does not translate into any polypeptide and undergo NMD (Fig. 7b). The longer transcript was preferentially detected upon RNF113A deficiency in Cisplatin-treated cells, indicating that the spliceosome removes intron 1 to generate the NUPR1-encoding transcript. Moreover, SF3B2-deficient cells also failed to induce *NUPR1* expression upon Cisplatin treatment (Supplementary Fig. 9). Conversely, RNF113A overexpression in A549 cells enhanced *NUPR1* expression after Cisplatin treatment (Fig. 7c). We also analyzed TCGA data and observed that both *RNF113A* and *NUPR1* expression were indeed positively correlated in lung cancer (Fig. 7d). Therefore, RNF113A, and by extension the spliceosome, promotes *NUPR1* expression upon DNA damage through the proper splicing of its transcript. In

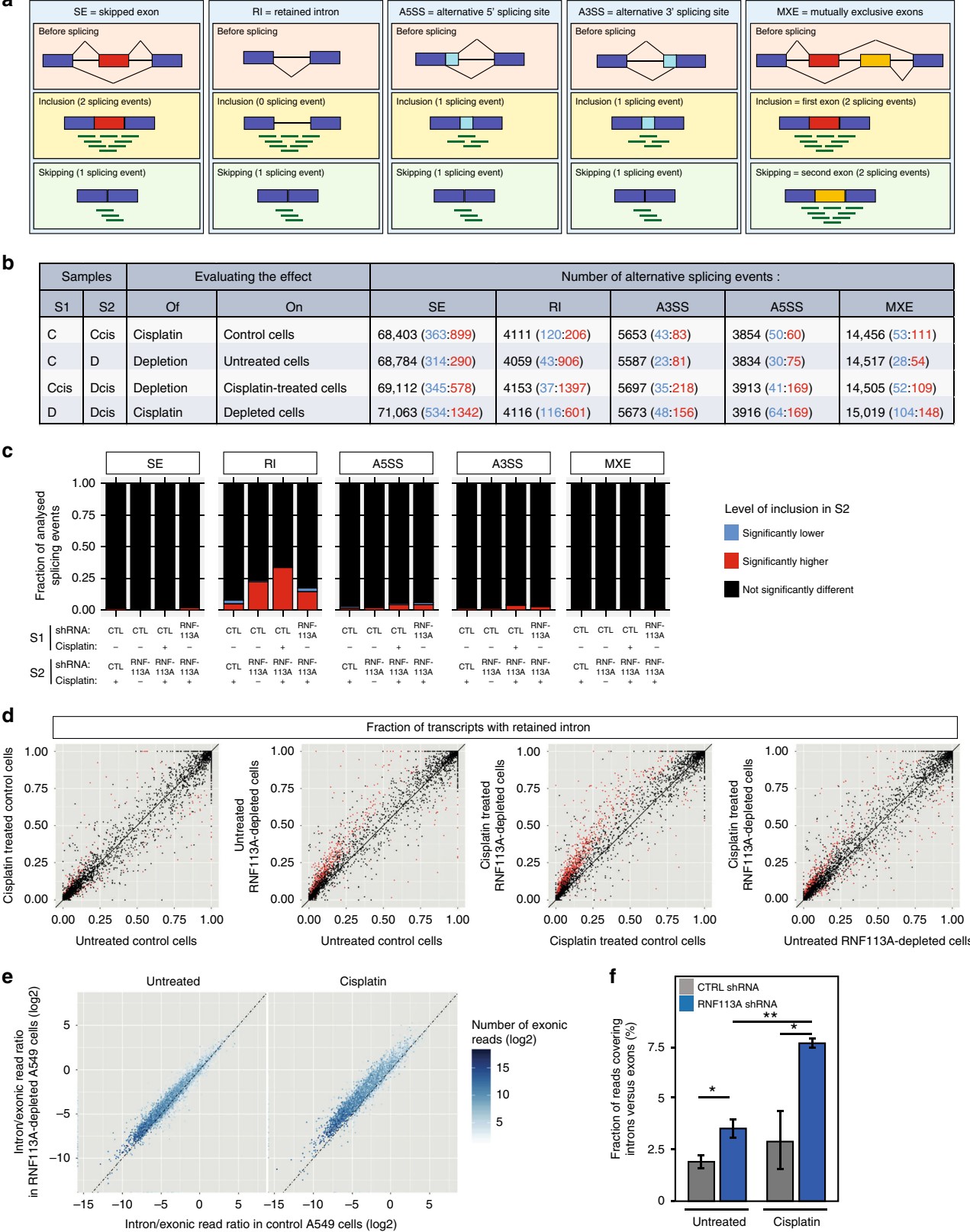

agreement with the fact that the genetic inactivation of *NUPR1* triggers senescence[43], the depletion of RNF113A in A549 cells led to senescence, as evidenced by an increase of β-galactosidase-positive cells (Fig. 7e). Therefore, RNF113A expression prevents senescence, at least by maintaining *NUPR1* expression in lung cancer cells.

The NADPH oxidase activator 1 Noxa1 catalyses the production of intracellular and extracellular superoxide ($O_2^-$) from $O_2$ and NADPH[44]. Interestingly, multiple transcripts of Noxa1 were detected in A549 cells (Fig. 8a). The 483 amino acids long Noxa1 protein is generated from the translation of a transcript lacking both introns 9 and 10 (Fig. 8a). Two additional transcripts, which

**Fig. 5 Increased intron retention in mRNAs upon RNF113A deficiency, especially in Cisplatin-treated lung cancer cells. a–d** Analysis of alternative splicing (AS) at the level of individual splicing events with rMATS (replicate Multivariate Analysis of Transcript Splicing). **a** Schematic representation of the different types of AS events analyzed by rMATS. Inclusion and skipping forms are quantified using reads overlapping junctions and reads unique to the inclusion form. **b** Number of AS events found in RNF113A-depleted and control A549 cells treated or not with Cisplatin. Each entry in the table has three values: Total number of events evaluated in that comparison (Inclusion significantly lower in S2: Inclusion significantly higher in S2). Samples are as follow: C = control shRNA—no drug, Ccis = control shRNA + Cisplatin, D = RNF113A shRNA—no drug, Dcis = RNF113A shRNA + Cisplatin. Significant events are defined as FDR<5% and delta inclusion level ($|\Delta\Psi|$) of at least 20%. **c** Fraction of analyzed AS events with inclusion level significantly lower or higher upon treatment (four comparisons) for each category of AS events. **d** Comparison of fraction of transcripts with retained intron in RNF113A-depleted and control A549 cells treated or not with Cisplatin. Each dot represents an intron retention event. Events with statistically significant difference between conditions are represented in red. Black broken line: identity axis. **e-f** Analysis of Intron Retention at the level of individual genes. **e** Comparison, for each protein-coding gene, of the ratio of mRNAseq reads in intronic versus exonic regions (average of three replicates) between RNF113A-depleted and control A549 cells treated or not with Cisplatin. Black broken line: identity axis. Shift towards higher ratios upon RNF113A depletion are significant ($p$-value $< 2.2 \times 10^{-16}$; Wilcoxon signed rank tests (paired tests); see Supplementary Fig. 8c). **f** Total number of reads covering introns versus total number of reads restricted to annotated exons for both RNF113A-depleted and control A549 cells treated or not with Cisplatin. Data were obtained from three replicates for each experimental condition (mean ± SD, two-sided $T$-test, **$p < 0.01$, **$p < 0.05$).

includes intron 9 and/or 10, were also detected and encode shorter 318 or 283 amino acids long Noxa1 proteins, respectively due to the appearance of premature stop codons (Fig. 8a). RNF113A deficiency had dramatic consequences on Noxa1 expression as the smaller 318 amino acids protein generated from the translation of the transcript which includes intron 10 ("Noxa1 ΔSH3") accumulated in RNF113A-depleted cells (Fig. 8b). This smaller isoform lacks the C-terminal SH3 domain and more efficiently activates NOX1[45] (Fig. 8a, b). The accumulation of Noxa1 ΔSH3 was also seen in SF3B2-depleted A549 cells (Fig. 8c). Consistently, the depletion of RNF113A in A549 cells potentiated ROS production in both unstimulated and Cisplatin-treated cells (Fig. 8d). Moreover, the NADPH/NADP ratio, which reflects NOX1 activity, was increased upon RNF113A deficiency in Cisplatin-treated cells (Fig. 8e). Therefore, RNF113A expression limits the production of ROS, at least through Noxa1 splicing.

**RNF113A stabilizes MCL-1**. ROS production destabilizes MCL-1[46]. Moreover, MCL-1 limits cell death triggered by Cisplatin in lung cancer cells[47]. Therefore, we investigated whether RNF113A controls MCL-1 expression. MCL-1 protein levels decreased in RNF113A-depleted A549, BZR-T33, H1975 or Calu-6 cells subjected to Cisplatin and undergoing Caspase 3-dependent apoptosis (Fig. 9a, Supplementary Fig. 10a, b and Fig. 9b, respectively). MCL-1 phosphorylation by GSK3 similarly decreased in RNF113A-depleted A549 cells, suggesting that a deregulation of MCL-1 phosphorylation, which triggers its degradation[48], was not responsible for MCL-1 disappearance from the cytoplasm upon Cisplatin treatment of RNF113A-depleted cells (Fig. 9a). On the other hand, RNF113A deficiency in Caspase 3-negative HCC827 cells did not impact on MCL-1 levels upon Cisplatin treatment but nevertheless triggered more Caspase 8 activation and more cell death (Fig. 9b and Supplementary Fig. 10c). On the other hand, RNF113A-overexpressing cells showed elevated MCL-1 protein levels when treated to Cisplatin and MCL-1 half-life was enhanced in Cisplatin-treated and RNF113A-overexpressing A549 cells (Supplementary Fig. 10d and Fig. 9c, respectively). MCL-1 stabilization was due to a defective proteasome-dependent degradation as the proteasome inhibitor MG132 restored MCL-1 protein levels in RNF113A-depleted A549 cells treated with Cisplatin (Fig. 9d). Moreover, polyubiquitin chains were also detected in a TUBE assay from RNF113A-depleted A549 cells treated with Cisplatin in which the proteasome was blocked (Fig. 9e). We next looked at expression levels of regulators of MCL-1 polyubiquitination and focused on USP9X, which deubiquitinates and stabilizes MCL-1[49]. USP9X levels were decreased in RNF113A-depleted

cells, which may contribute to the enhanced degradative polyubiquitination of MCL-1 (Fig. 9f). Our data has some clinical relevance as RNF113A and MCL-1 protein levels positively correlated in clinical cases of lung cancer (Fig. 9g). SF3B2-depleted cells, which express less RNF113A, also had less MCL-1, at least because of decreased USP9X levels (Fig. 9h).

Given the central role of BCL-2 in cell survival, BH3 mimetics/BCL-2 inhibitors such as ABT-737 were designed and indeed causes regression of established tumors[50]. Nevertheless, MCL-1 promotes resistance to ABT737[51]. As RNF113A stabilizes MCL-1, we reasoned that RNF113A deficiency may enhance ABT737-dependent cell death through MCL-1 downregulation. Indeed, cell death upon treatment with ABT737 was more pronounced upon RNF113A deficiency in A549 cells (Fig. 9i). Therefore, RNF113A acts as a pro-survival candidate, at least by maintaining MCL-1 protein levels.

**RNF113A deficiency circumvents resistance to Cisplatin**. As MCL-1 promotes resistance to Cisplatin and because MCL-1 expression relies on RNF113A, we next explored whether RNF113A also contributes to resistance to Cisplatin in lung cancer cells. We generated Cisplatin-resistant A549 cells by culturing parental A549 cells ("A549/P") with increasing concentrations of Cisplatin. Resistant cells ("A549/CR (4.5)") did not dramatically undergo cell death when subjected to increasing concentrations of Cisplatin and underwent epithelial to mesenchymal transition (EMT), as judged by an elongated morphology and by lower levels of E-cadherin (Fig. 10a). RNF113A deficiency sensitized these resistant cells to cell death triggered by Cisplatin (Fig. 10b). Mechanistically, RNF113A deficiency triggered Caspase 9 and 3 activation upon treatment with Cisplatin, at least due to enhanced DNA-PKcs phosphorylation and to decreased MCL-1 protein levels (Fig. 10c). The defective expression of SF3B2, Rad51, and RNF8 seen upon RNF113A deficiency in parental A549 cells was also observed in Cisplatin-resistant cells (Fig. 10d). As a result, cell death triggered by ABT737 was more pronounced upon RNF113A deficiency in Cisplatin-resistant A549 cells (Fig. 10e). RNF113A deficiency also triggered tumor regression in vivo upon treatment with Cisplatin when resistant A549 cells were transplanted into immunodeficient mice (Fig. 10f). Therefore, RNF113A promotes chemoresistance to Cisplatin in lung cancer cells, at least by stabilizing MCL-1 levels.

## Discussion

We show here that RNF113A promotes cell survival upon DNA damage as a spliceosome subunit. Splicing targets of

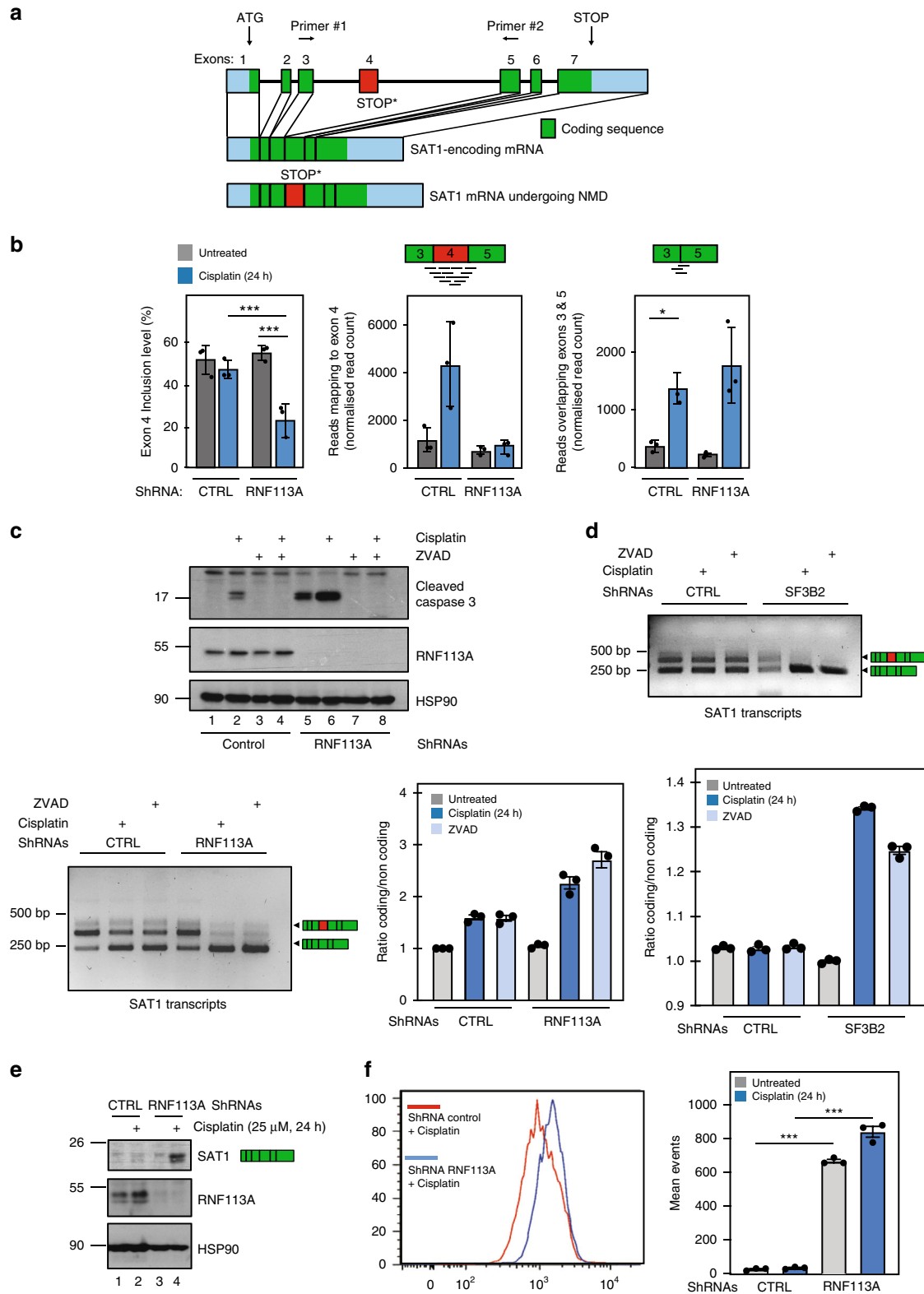

RNF113A include multiple pro-survival candidates. Interfering with RNF113A triggers cell apoptosis, at least through MCL-1 destabilization, which circumvents the acquired resistance to BCL-2 inhibitors. RNF113A deficiency also triggers ferroptosis, at least through SAT1 expression and enhances ROS production upon DNA damage. SF3B2 deficiency leads to very similar consequences as the loss of RNF113A, which suggests that both proteins have similar functions. Therefore, the spliceosome is a major actor of cell survival in lung cancer and also define RNF113A as a promising anti-cancer target to fight the acquired resistance to BCL-2 inhibitors.

The mechanisms by which RNF113A promotes MCL-1 stability may be cell-type dependent. Indeed, Cisplatin-resistant cells depleted for RNF113A show appearance of the short and

**Fig. 6 RNF113A regulates the splicing of SAT1. a–c** RNF113A promotes *SAT1* splicing. **a** Representation of *SAT1* transcripts. The coding and non-coding sequences are illustrated as green and blue rectangles, respectively. A defective splicing of exon 4 (red rectangle) leads to a premature STOP codon. **b** rMATS results for the analysis of exon 4 skipping in *SAT1* transcripts. The first barplot on the left represents the inclusion level of exon 4 in *SAT1* transcripts in RNF113A-depleted or control cells, treated or not with Cisplatin. The middle and right plots represent the number of reads specific to transcripts with or without exon 4, respectively. Read counts have been normalized for differences in sample sequencing depth using DESeq2 size factor (median ratio method). Data were obtained from 3 replicates for each experimental condition (mean ± SD; Left panel: *** = FDR *q*-value < 0.001 from rMATS; Middle and right panels: *$p < 0.05$, two-sided *T*-test on normalized read counts). **c, d** RNF113A and SF3B2 deficiencies share common defects in SAT1 splicing. Transcripts are detected by RT-PCR experiments, using both primers depicted in a. Data for both control and RNF113A or SF3B2-depleted cells are shown (**c** and **d**, respectively). A quantification of all signals is illustrated for both experiments. On the top (**c**), WB analyses were done with control or RNF113A-depleted cells treated or not with Cisplatin (25μM for 24 h) and treated or not with ZVAD (20 μM for 24 h). **e** Enhanced SAT1 protein levels in Cisplatin-treated and RNF113A-deficient lung cancer cells. Control and RNF113A-depleted A549 cells were treated or not with Cisplatin (25 μM for 24 h) and WB analyses were done. **f** RNF113A deficiency triggers ferroptosis upon Cisplatin stimulation. Control and RNF113A-depleted A549 cells were treated or not with Cisplatin (25 μM) and lipid ROS were quantified by BODIPY-C11 staining, using a flow cytometer. Data from two experiments performed in triplicates are illustrated (mean ± SD) (***$p < 0.001$, Student *t*-test).

pro-apoptotic form of MCL-1 and lower levels of the long and pro-survival form of MCL-1, suggesting that MCL-1 is a direct target of the spliceosome in these cells. We did not find any evidence that MCL-1 splicing was regulated by RNF113A in parental cells. Yet, MCL-1 degradative polyubiquitination was enhanced upon RNF113A deficiency in these cells. USP9X, which stabilizes MCL-1 by promoting its deubiquitination[49], is less expressed upon RNF113A deficiency. This mechanism may contribute to the destabilization of MCL-1 seen in RNF113A-depleted cells. Alternatively RNF113A may directly inhibit the function of any E3 ligase such as MULE that targets MCL-1 for degradation upon DNA damage[52]. Another molecular mechanism may involve ROS production. Indeed, ROS, which are more produced upon RNF113A deficiency, destabilize MCL-1 through a poorly characterized pathway[46,53].

RNF113A deficiency leads to multiple types of cell death in addition to apoptosis upon DNA damage. Indeed, Caspase 3-deficient lung cancer cells still undergo cell death upon RNF113A deficiency when treated with Cisplatin. We actually show that ferroptosis occurs when RNF113A-depleted cells are subjected to a DNA damage signal.

The repair of DNA alkylation damage involves the alkylation repair complex ASCC (activating signal cointegrator complex) which relocalizes to specific nuclear foci with spliceosome proteins and basal transcription factors upon exposure to alkylating agents[54]. This recruitment to nuclear foci requires the sensing of polyubiquitin chains by the CUE (coupling of ubiquitin conjugation to ER degradation) of ASCC2. RNF113A is the E3 ligase that catalyses the formation of these K63-linked non degradative polyubiquitination chains on BRR2, an ASCC2-interacting protein[54]. RNF113A is relocalized to ASCC2-enriched foci upon DNA damage, which appears to be distinct from pH$_2$A.X S139$^+$ foci, a finding that we also report in our study. The N-terminal domain of RNF113A is critical for the binding to BRR2. We show here that this domain includes the NLS of RNF113A, which is consistent with the idea that all functions of RNF113A in the nucleus critically relies on its N-terminal domain. Interestingly, we demonstrate that Cisplatin triggers the cytoplasmic shuttling of RNF113A. SF3B2 also disengages from the chromatin upon DNA damage. This observation is not in contradiction with the fact that splicing events occur in the nucleus as a pool of both nuclear RNF113A and SF3B2 can still be found in lung cancer cells showing some DNA damage. Spliceosome subunits such as SR proteins are actually dephosphorylated to facilitate the export of spliced mRNAs to the cytosol in order to enhance translation[55]. Whether post-translational modifications of RNF113A regulates its cellular localization remains unknown. In any case, this strongly suggests that RNF113A moves into the cytoplasm, presumably with some spliced transcripts upon DNA damage.

Let's note however that the pool of cytoplasmic SF3B2 did not increase upon Cisplatin treatment, which differs from RNF113A. This is the only property that distinguishes SF3B2 from RNF113A.

It is unclear whether RNF113A exclusively works as a spliceosome subunit. As SF3B2 deficiency mimics the phenotypical alterations seen in RNF113A-depleted cells, this suggests that RNF113A works as a spliceosome subunit. Yet, the fact that RNF113A polyubiquitinates proteins such as BRR2, which does not regulate RNA splicing[4], indicates that some spliceosome-independent functions of RNF113A may also occur. Moreover, the co-localization of RNF113A with DNA-PKcs in Cisplatin-treated A549 cells and the defective engagement of pH$_2$AX to DSB sites in depleted DIvA cells suggests its recruitment at the extremities DNA DSBs where it could act as an E3 ligase to directly promote DNA repair. Alternatively, RNF113A may indirectly promote this process as a spliceosome subunit. We actually demonstrate that RNF8, which promotes histone polyubiquitination and the recruitment of 53BP1 and BRCA1 repair proteins to double-strand breaks[56], is a target of RNF113A as a spliceosome subunit. Therefore, RNF113A is involved in DNA repair through both direct and indirect mechanisms. The absence of strong ATR activation observed in RNF113A-depleted cells allowed to resume growth in a Cisplatin-free media, indicates either a deficient replication-stress response or more likely the persistence of damage-induced cell cycle arrest due to the high level of damage present. All experiments were conducted after 24 h of permanent contact with Cisplatin. Only one experiment was performed with a recovery period in order to best visualize ATR activation, which requires replication.

Our data demonstrate that the spliceosome contributes to the acquired resistance to Cisplatin, at least by promoting MCL-1 stability. Therefore, targeting RNF113A or SF3B2 is a strategy to circumvent the acquired resistance of lung cancer cells to Cisplatin. Patients suffering from lung cancer and showing some acquired resistance to ABT737 may also benefit from the inhibition of the spliceosome. To conclude, it is tempting to speculate that any tumors showing some MCL-1 stabilization may benefit from spliceosome inhibitors targeting RNF113A or SF3B2.

## Methods

**Cell lines, antibodies, plasmids, and treatments**. The human adenocarcinoma A549 (CCL-185) cell line, H1975 (CRL-5908) and 293 (CRL-1573) cells as well as normal human dermal fibroblasts (PCS-201-012) were purchased from the American Type Culture Collection (ATCC, Manassas, VA, USA). BZR-T33 cells were kindly provided by Dr. Christine Gilles (GIGA-Cancer, University of Liege, Belgium). The Lenti-X 293T cell line was obtained from Clontech Laboratories (catalog number 632180) (Palo Alto, CA, USA). All cell lines (including normal human dermal fibroblasts) were tested for mycoplasma contamination and were maintained in Dulbecco's Modified Eagle's Medium (DMEM) (Lonza, Basel,

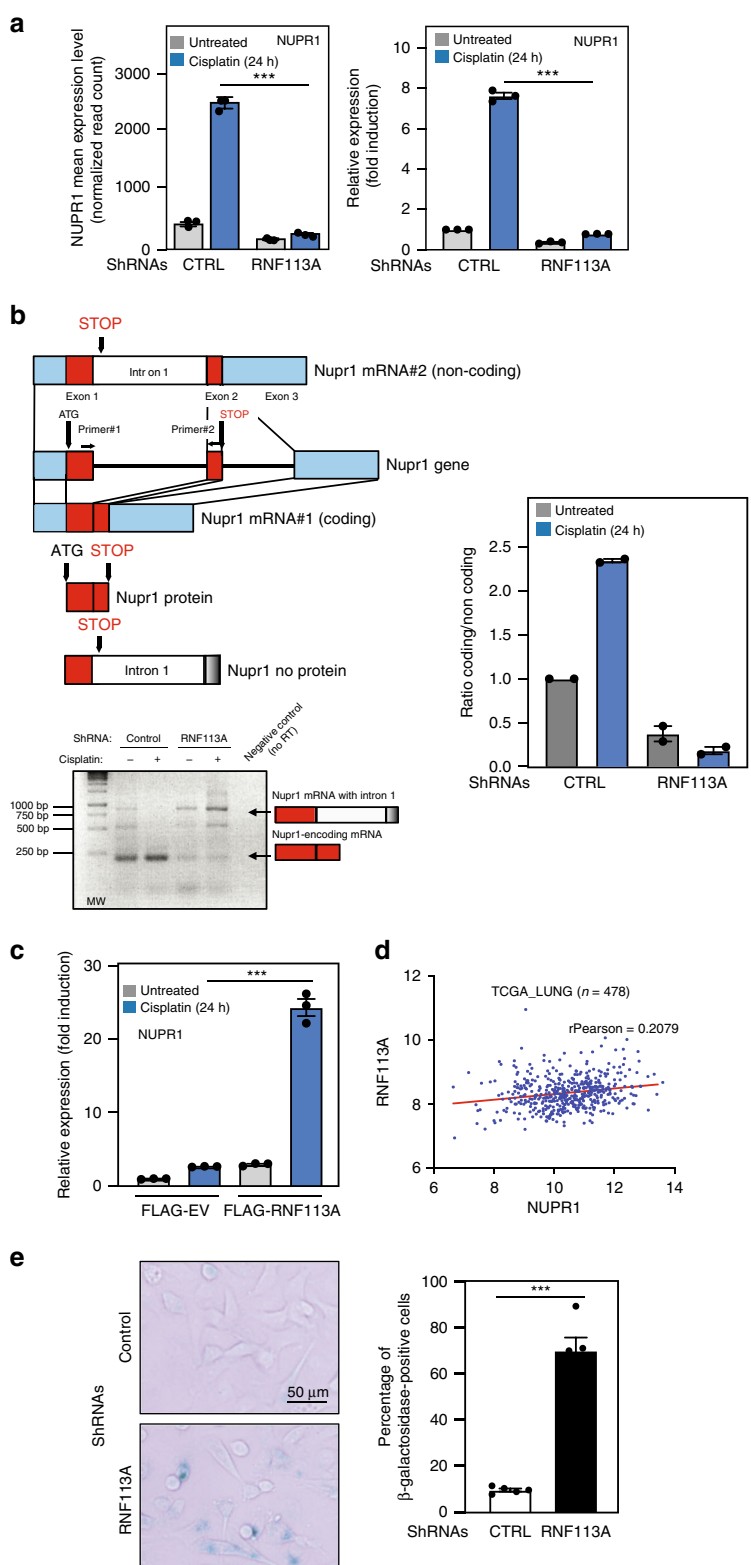

Switzerland) supplemented with 10% Fetal Bovine Serum (FBS) (Sigma-Aldrich, St-Louis, MO, USA), L-glutamine and antibiotics (Lonza). ZVAD was purchased from Promega (Madison, WI). A list of antibodies and primers used in this study are provided in the Supplementary Tables 1 and 2, respectively. A polyclonal anti-RNF113A antibody was raised in rabbit and was directed against a peptide derived from the C-terminal region of human antigen (Phoenix Pharmaceuticals, Burlingame, CA, USA) and was used for endogenous immunoprecipitations. MG132 was from A&E Scientific (Marcq, Belgium). Cisplatin (cis-

Diammineplatinum(II) dichloride), Camptothecin, and Etoposide were obtained from Sigma-Aldrich. Cisplatin was freshly prepared before each experiment in 0.15 M NaCl. Camptothecin and Etoposide were prepared in DMSO at 10 mM or 25 mM and stored at −20 °C. The FLAG-RNF113A construct was generated by subcloning the corresponding PCR amplified cDNA from A549 cells into the pcDNA3.1 construct (Clontech). For stable transfections, wild type RNF113A was subcloned from the pcDNA3.1 plasmid into the pIRES puromycin vector (Clontech).

**Fig. 7 RNF113A controls NUPR1 expression. a** RNF113A promotes Cisplatin-induced NUPR1 expression. The graph on the left represents the mean normalized number of reads mapping to *NUPR1* gene in each experimental condition (mean ± S.D.; adjusted *p*-value from DESeq2 analysis of 3E-19 for the effect of RNF113A depletion, 9E-10 for cisplatin treatment and 4E−26 for the interaction between depletion and treatment, Supplementary Data 2). The graph on the right represents the level of *NUPR1* mRNA in the same conditions. *NUPR1* mRNA levels in unstimulated cells is set to 1 and levels in other experimental conditions are relative to that after normalization with β-actin. Data from two independent experiments performed in triplicates (means ± SD) (\*\*\**p* < 0.001, Student *t*-test). **b** RNF113A promotes the splicing of intron 1 from *NUPR1* mRNA, especially upon Cisplatin treatment. A first transcript has a STOP codon within exon 2 and a second transcript includes intron 1. This transcript does not code for any protein due to a premature STOP codon within intron 1. At the bottom, RT-PCR experiments were done. The shorter and NUPR1-encoding transcript is preferentially detected in control cells. A quantification of all signals is illustrated on the right. **c** RNF113A promotes the expression of NUPR1-encoding mRNA. Real-Time PCRs were carried out with extracts from control versus RNF113A-overexpressing A549 cells treated or not with Cisplatin and mRNA levels of the NUPR1-encoding transcript were quantified in all experimental conditions. *NUPR1* mRNA levels are plotted as in (**a**). Data are from two independent experiments performed in triplicates (means ± SD) (\*\*\**p* < 0.001, Student *t*-test). **d** *RNF113A* and *NUPR1* expression are positively regulated in lung cancer (TCGA analyse, *n* = 478). **e** RNF113A-depleted cells undergo senescence. β-galactosidase staining in control or RNF113A-depleted A549 cells are shown. The percentage of β-galactosidase positive cells in each population is presented (means ± SD) (\*\**p* < 0.01, Student *t*-test).

**Generation of Cisplatin-resistant lung cancer cell lines**. The lung cancer-derived cell line A549 was used as the parental line (A549/P) from which was generated the Cisplatin-resistant cell line (A549/CR). The A549/P cell line was serially subcultured through incrementally increasing Cisplatin concentrations (from 0.5 to 5 μM for A549/P) for up to five months. The A549/CR cell line retained the capacity for proliferation when returned to medium containing 4.5 μM Cisplatin.

**Patient samples and Tissue Micro Array (TMA) analyses**. A TMA of 40 formalin-fixed, paraffin-embedded lung adenocarcinomas was immunohistochemically stained for RNF113A. The TMA consisted of 12 KRAS-mutated AC (11 in Exon 2, 1 in Exon 3), 5 EGFR mutated AC (3 in Exon 19, 2 in Exon 21) and 23 KRAS-/EGFR-wildtpe adenocarcinomas. Immunohistochemical staining (IHC) was performed on an automated stainer (Thermo Lab Vision 480S) using the anti-RNF113A-antibody (#HPA000160, Sigma-Aldrich). Normal lung from three patients and lung cancer tissues of patients suffering from adenocarcinomas were collected at the University Hospital of Cologne and were subjected to IHC and western blot analyses. Informed consent was obtained from these patients. The TMA-slide was dewaxed, pretreated in citrate buffer at pH6 for 20 min and then incubated with the anti-RNF113A antibody for 30 min. Following an H²O²-Block (Thermo scientific TA-125-HP), the slide was treated with enhancer (10 min; Immuno Logic c-DPVB blocking), polymer (15 min; Immuno Logic c-DPVB999HRP) and DAB (8 min; Immuno Logic BS04-999A + BS04-999B) with interjacent buffer wash (Thermo scientific TA-999-TT). Finally, the slide was counterstained with hematoxylin. Nuclear and cytoplasmic staining of tumor cells was regarded as positive. Staining intensity was scored as weak, moderate or strong and the percentage of positive tumor cells was estimated. Stromal cells served as internal negative control.

**Transfection and lentiviral infections**. Transient and stable transfections were performed using the Mirus TransIT-LT1 transfection reagent (Mirus Bio, Madison, WI, USA). For lentivirus-mediated shRNA experiments in BZR-T33, A549, H1975, HCC827, Calu-6 and DIvA cells, $3 \times 10^6$ 293-LentiX cells (Clontech) were transfected with 12 μg of the "non target" lentiviral shRNA plasmid (used as negative control) or with the shRNA construct that targets human RNF113A, 12 μg of psPAX2 and 5 μg of VSVG plasmid, using the Mirus Bio's TransIT-LT1 reagent. The supernatants of those infected cells were collected and filtered (0.2 μm) 48 h after transfection and added with polybrene (5 μg/ml) to $3 \times 10^6$ target cells. This latter step was repeated once after 24 h. Depleted cells were selected and maintained in normal culture medium supplemented with 1 μg/ml puromycin (InvivoGen, Toulouse, France).

**Cell lysis, cytoplasmic and nuclear fractionation**. Total cell extracts were obtained by washing cells in PBS before lysis in 1% SDS lysis buffer. Extracts were heated at 95 °C for 5 min. For the isolation of cytoplasmic and nuclear fractionations, cells were harvested and lysed in the Cytoplasmic Lysis Buffer (CLB) (10 mM Tris-HCl pH 7.9, 340 mM sucrose, 3 mM $CaCl_2$, 0.1 mM EDTA, 2 mM $MgCl_2$, 1 mM DTT, 0.5% NP-40) supplemented with cOmplete Protease Inhibitor (Roche). After centrifugation, the supernatant was separated and centrifuged again to obtain the cytoplasmic fraction. For western blot analyses, the pellet was lysed in 1% SDS lysis buffer to obtain the nuclear fraction. Uncropped and unprocessed scans generated from all western blot analyses illustrated in this study are provided as a Supplementary Data.

**Immunoprecipitations of polyubiquitinated proteins**. Endogenous immunoprecipitations of ubiquitinated proteins were performed with control TUBE, TUBE 1 or 2 agarose (LifeSensors, Malvern, PA, USA) according to the manufacturer's instructions in a buffer containing 50 mM Tris-HCl pH 7.5, 150 mM NaCl, 1 mM EDTA, 1 mM $Na_3VO_4$, 10 mM sodium glycerophosphate, 50 mM NaF, 1% NP40, protease inhibitor (Roche). TUBE control agarose was used as negative control.

**ChIP assays**. ChIP assays were carried out with extracts from A549 cells left untreated or stimulated with Cisplatin (25 μM for 24 h) as well as with extracts from DIvA U2-OS cells (stably expressing ASiSI-ER) (a gift from Gaëlle Legube, University of Toulouse, Toulouse, France) left untreated or stimulated with 4-hydroxy Tamoxifen (300 nm for 4 h). For experiments carried out with AsiSI-ER-U2OS cells, control or Tamoxifen-treated cells and A549 cells treated or not with Cisplatin were crosslinked for 10 min at room temperature (1/10th of the cross-linking mix was directly added to the plate containing the cells and the culture medium). The reaction was quenched by adding 1/10th the volume of 1.25 M glycine to reach 125 mM of concentration. Cells were then washed twice with PBS and lysed with the lysis buffer (1% SDS, 10 mM EDTA, pH8.0, 50 mM Tris-Hcl, pH 8.0 with protease inhibitors). Cells were harvested by scraping from the plates. Cell lysates were sonicated on ice for 15 min using the Bioruptor sonicator (Diagenode, Liege, Belgium). Lysates were then spin down for 5 min at maximum speed in a bench-top centrifuge and diluted 10 times with the dilution buffer (1% Triton X-100, 150 mM NaCl, 2 mM EDTA, pH 8.0, 20 mM Tris-HCl, pH 8.0 with protease inhibitors). Five micrograms of antibody was added for each experimental condition and tubes were rotated overnight at 4 °C. Simultaneously, protein A or G beads were washed with the dilution buffer and pre-absorbed with 100 μg/ml BSA in rotating tubes overnight at 4 °C. The next day, beads were added to the Immunoprecipitation (IP) mix and rotated for 1–2 h at 4 °C. All beads were subsequently washed three times with the washing buffer (1% Triton X-100, 0.1% SDS, 150 mM NaCl, 2 mM EDTA, pH 8.0, 20 mM Tris-HCl, pH 8.0 with protease inhibitors), once with the final washing buffer (1% Triton X-100, 0.1% SDS, 500 mM NaCl, 2 mM EDTA, pH 8.0, 20 mM Tris-HCl, pH 8.0 with protease inhibitors) and once with the washing buffer B (20 mM Tris pH 8.0, 1 mM EDTA, 250 mM LiCl, 0.5% NP-40, 0.5% Na-deoxycholate, protease inhibitors). Four hundred and fifty microliters of elution buffer (1% SDS, 100 mM $NaHCO_3$) were added to each IP reaction. For inputs, 400 μl of elution buffer were added to 50 μl of each lysate. Proteinase K and RNase A (500 μg/ml each) were added to each reaction and incubated for 30 min at 37 °C. Samples were subsequently reverse cross-linked at 65 °C overnight (20 μl of 5 M NaCl were added prior the reverse crosslinking to each reaction). DNAs were purified using the phenol-chloroform extraction method, precipitated with ethanol and resuspended in 100 μl of sterile water. 1 μl was used for PCR reactions.

**Immunofluorescence**. Cells were seeded on coverslips in 6-well plates. After transfection or treatment, cells were washed in PBS, fixed with 4% PFA in PBS for 15 min or in ice cold methanol for 20 min on ice (for hybrids) and subsequently pre-immobilized with 0.2% Triton-X100/PBS for 10 min at room temperature. Cells were then washed and blocked for 1 h (5% BSA or 5% normal goat serum in PBS) followed by overnight incubation at 4 °C with primary antibodies in a blocking solution. The next day, coverslips were incubated for 1 h with appropriate goat secondary antibodies coupled to Alexa Fluor 488 or 568 fluorophores (Life Technologies), washed and incubated for 10 min with DAPI (Life Technologies). ProLong (Life Technologies) was used for mounting on glass slides and images were acquired with the Leica TCS SP5 II confocal system (Leica Microsystems, Wetzlar, Germany). For pre-extraction in the A549 cell line, cells were washed with PBS and incubated twice for 3 min with the CSK buffer (10 mM PIPES pH 7.0, 0.1% Triton-X100, 100 mM NaCl, 300 mM sucrose, 3 mM $MgCl_2$) supplemented with 0.3 mg/ml RNase A (Roche). Cells were washed in PBS and fixed with 2% PFA/PBS. For the quantification of $pH_2AX$ (S139) foci and pDNA-PKcs (Serine 2056) positive cells, at least 10 random fields with approximately 100 cells were manually counted.

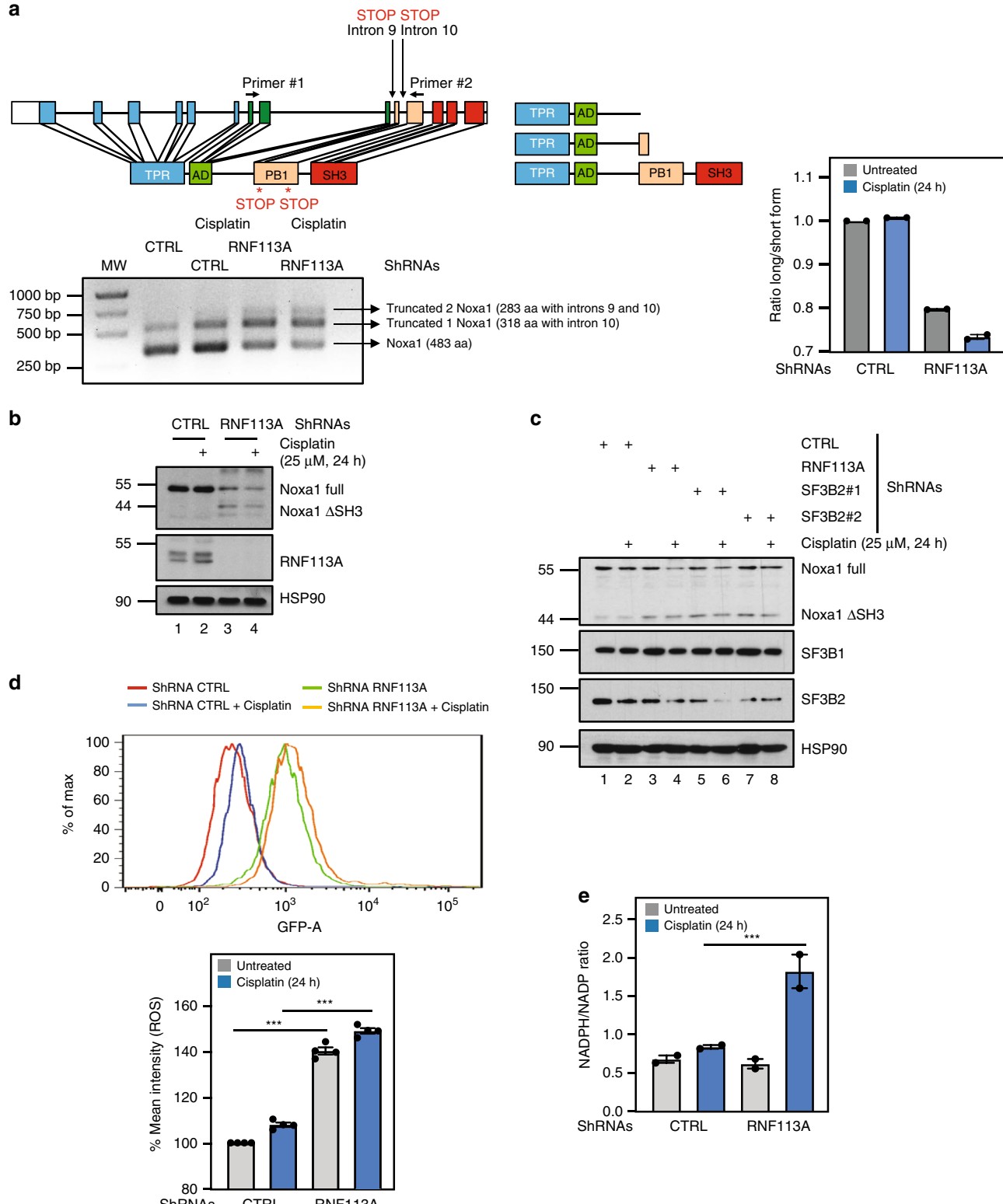

**Real-time PCRs**. Total RNAs were extracted from cells in triplicates using the NucleoSpin RNA extraction kit (Macherey-Nagel, Düren, Germany) that provides rDNase I treatment to ensure the digestion of DNA in the sample. cDNAs were synthesized using the Revert Aid™ H Minus First Strand cDNA Synthesis kit (Fermentas, Glen Burnie, MD, USA) and PCRs were performed using the Power SYBR Green PCR Master kit (Applied Biosystems, Foster City, CA, USA) on the LightCycler 480 (Roche). The primers amplifying human β-actin (5′-GCTACGA GCTGCCTGACG and 3′-GGCTGGAAGAGTGCCTCA) were used for normalization, and the primers 5′-AGGCGGTGGATCAGGTGTGC and 3′-ACAGTGC AGCCTTCGTCGCT were used for the amplification of RNF113A.

**Comet assay**. Control and RNF113A-depleted BZR-T33 cells were treated with 25 μM Cisplatin for 20 h. 7500 cells were suspended in 0.75% LMP agarose and spread on 20-well CometSlides (Trevigen, Gaithersburg, MD, USA). Cells were lysed for 2 h at 4 °C in a buffer containing 2.5 M NaCl, 100 mM EDTA, 1% Triton X-100, and 10 mM Tris, pH 10. Slides were placed in an electrophoresis unit and DNA was allowed to unwind for 40 min in the running buffer (300 mM NaOH, 1 mM EDTA, pH > 13). Electrophoresis was conducted for 30 min at 0.77 V/cm. The slides were neutralized with 0.4 M Tris, pH 7.5, stained with 2 mg/ml DAPI and covered with cover slips. A total of 50 images were randomly taken from each sample using the Leica TCS SP5 II confocal system and the Olive tail moment was

**Fig. 8 RNF113A limits the production of ROS through Noxa1 expression upon DNA damage. a** RNF113A controls the splicing of the Noxa1 transcript. A shematic representation of the Noxa1 gene is illustrated. Three distinct Noxa1 transcripts are detected by RT-PCR using primers 1 and 2 located within exon 7 and 11, respectively. A first transcript, named "Truncated 2 Noxa1" includes introns 9 and 10 and gives rise to a 283 amino acids long Noxa1 protein. A second transcript referred to as "Truncated 1 Noxa1" has intron 10 and gives rise to a 318 amino acids long protein whereas a third transcript lacking both introns 9 and 10 codes for a 483 amino acids long Noxa1 protein. A quantification of all signals is illustrated on the right. **b**, **c** RNF113A or SF3B2 deficiency interferes with Noxa1 protein synthesis. Control (**b**, **c**), RNF113A (**b**)- or SF3B2 (**c**)-depleted A549 cells treated or not with Cisplatin (25 μM for 24 h) were subjected to WB analyses. **d** RNF113A limits ROS production upon DNA damage by regulating Noxa1 splicing. Control or RNF113A-depleted A549 cells were treated or not with Cisplatin (25 μM for 24 h) and ROS production was accessed using carboxy-H2DFFDA. Cells were analyzed on FACS Canto II and the data were generated using the FlowJo program. Representative plot from one of multiple experiments shows an increase of ROS production in RNF113A-depleted cells which further increases after Cisplatin treatment (on the top). At the bottom, data are illustrated in the histogram build using % of mean intensity received after analysis data using the BD FACSDIVA software. Data were obtained with three replicates for each experimental condition from three independant experiments (mean ± SD, two-sided $T$-test (***$p < 0.001$). **e** Increased NADPH/NADP ratio upon RNF113A deficiency after treatment with Cisplatin (25 μM for 24 h). The NADPH/NADP ratio in control and RNF113A-depleted cells treated or not with Cisplatin was established (see the "Methods" section for details). Results of two independent experiments are shown (***$p < 0.001$).

calculated using the Comet Assay VI image analysis system (Perspective Instruments, Bury St Edmunds, UK).

**MTS assays**. For MTS assays, A549/P and A549/CR cell lines (1500 cells per well), counted by the TC20™ Automated Cell Counter (Bio-Rad, Pleasanton, CA, USA) were plated in 96-well flat bottom plates in triplicate and then treated with various concentrations of Cisplatin (0, 1, 3, 5, 10, 30, 50, and 100 μM) for 72 h. Cell viability was determined using the MTS assay reagent (CellTiter 96 AQueous one Solution Cell proliferation Assay; Promega, Madison, WI, USA) according to the manufacturer's protocol. The absorbance was measured at 490 nm using a Wallac Victor[2] 1420 Multilabel counter (Perkin Elmer, Wellesley, MA). Absorbance of untreated cells was designated as 100% and the number of viable cells in other experimental conditions were relative to that.

**Oligo (dT) pulldowns**. These pulldowns were carried out as described with A549 rather than HeLa cells[57].

**Apoptosis and ferroptosis assays and ROS quantification**. Control or RNF113A-depleted A549 cells, as well as control or RNF113A-overexpressing cells, were untreated or treated with 25 μM Cisplatin for 20 h. The Annexin-V-FLUOS Staining Kit (Roche) was used to access cell death following the manufacturer's instructions, using the FACSCalibur flow cytometer (BD Biosciences).

For the measurement of lipid peroxidation, control or RNF113A-depleted A549 cells treated or not with Cisplatin (25 μM for 24 h) were incubated for 30 min at 37 °C in a medium supplemented with 5 μM C-11 BODIPY (Invitrogen). The medium was then removed. Cells were trypsinized and centrifuged at 1500 rpm for 5 min at room temperature. The supernatant was subsequently removed and the pellet was washed in PBS once and resuspended in 100 μl of PBS. Cells were analyzed on FACS Canto II and the data were generated using the FlowJo program. Upon oxidation, the red emitting reduced form of the dye (595 nm) is converted into a green emitting oxidized form (520 nm).

For ROS measurement, control or RNF113A-depleted A549 cells treated or not with Cisplatin (25 μM for 24 h) were incubated for 10 min at 37 °C in a carboxy-H2DFFDA-containing solution at a concentration of 24 μM (Invitrogen). The medium was then removed. Cells were trypsinized and centrifuged at 1500 rpm for 5 min at room temperature. The supernatant was subsequently removed and the pellet was washed in PBS once and resuspended in 100 μl of PBS. Cells were analyzed on FACS Canto II and the data were generated using the FlowJo program. For all FACS analyses, the gating strategy used was forward and side scatter gating to remove debris and other events of non-interest while preserving cells based on size and or complexity. This gating strategy is illustrated in Supplementary Fig. 11.

**Measurement of senescence**. Senescence of control or RNF113A-depleted A549 cells was assessed using the Senescence β-Galactosidase Staining Kit (#9860) from Cell Signaling.

**Establishment of the NADPH/NADP ratio**. The NADP/NADPH ratio in control or RNF113A-depleted A549 cells treated or not with Cisplatin (25 μM for 24 h) was established using the NADP/NADPH Quantification Kit (MAK038) provided by Sigma.

**High-throughput RNA sequencing, pre-processing, and mapping**. RNA sequencing was performed on libraries prepared from total RNA samples. Three biological replicates were analyzed for each condition. RNA integrity was verified on a Bioanalyser 2100 with RNA 6000 Nano chips (Agilent technologies, CA, USA). RNA integrity number score was above 8 for every sample. Libraries were prepared using Truseq® stranded mRNA Sample Preparation Kits (Illumina, CA, USA) following manufacturer's instructions. Libraries were validated on the

Bioanalyser DNA 1000 chip and quantified by qPCR using the KAPA library quantification kit. Libraries were multiplexed and sequenced in two runs on an Illumina NextSeq500 sequencer to generate more than ~45,000,000 paired-end reads (2 × 76 bases) per library. Raw reads were demultiplexed and adapter-trimmed using Illumina bcl2fastq conversion software v2.17. After removal of reads aligning to rRNA, tRNA or mitochondrial sequences, the remaining reads were trimmed to 73 nucleotides to enable compatibility with rMATS v 3.2.5[41]. Reads were aligned to the human hg19 reference genome (downloaded with annotations from Illumina iGenomes website) using STAR v2.5.2b[58]. Quality of the sequencing data was successfully controlled using FastQC (http://www.bioinformatics.babraham.ac.uk/projects/fastqc/) and Picard tools (http://broadinstitute.github.io/picard/) along with MultiQC[59].

**Differential expression analysis at the gene level**. For the differential expression analysis at the gene level, raw gene counts were generated using STAR for all annotated transcripts. Differential expression analysis was performed using the DESeq2 R package[60] comparing either samples treated with Cisplatin to samples treated with no drug or samples treated with shRNF113A to samples treated with shCtrl in a model including both factors (additive model: "Expression level ~ Drug + Depletion"). Genes for which the effect of depletion was different depending on Cisplatin treatment (i.e., deviance from a purely additive contribution of the two factors) were also identified by performing a Wald test on the interaction term of a model "Expression level ~ Drug + Depletion + Drug * Depletion" (referred to as "Complex patterns" in the manuscript). In this interaction test, the following ratio is assessed: (shRNF113A + Cisplatin/shRNF113A—no drug)/(shCtrl + Cisplatin/shCtrl—no drug). Adjustment of the p-values to correct for multiple tests was performed using the Benjamini-Hochberg procedure. Genes with a FDR $q$-value below 0.05 were considered significantly affected and were further filtered based on the magnitude of the effect (difference in expression between conditions of at least twofold). In Supplementary Fig. 7a and Supplementary Data 2, log2 fold changes for "shRNF113A versus shCtrl" and for "Cisplatin versus no drug" are corrected for the over-dispersion due to low counts using DESeq2 shrinkage procedure. Gene set enrichment analyses were performed with the GSEA software[61], using shrunk log2 fold changes from DESeq2 as ranking metric.

**Alternative splicing analysis with rMATS**. Alternative splicing analyses were performed on the same STAR alignments as above using rMATS (replicate Multivariate Analysis of Transcript Splicing) v3.2.5[41]. In rMATS, splicing events are analyzed individually and by categories of events (Skipped exons, mutually exclusive exons, retained intron, alternative 5′ splicing site and alternative 3′ splicing site). For this analysis, Ensembl annotations (GRCh37 release 75) were used because they contain annotations for several retained introns. This allowed rMATS to detect and analyze up to 4,153 intron retention events. Inclusion and skipping forms were quantified using reads overlapping junctions and reads unique to the inclusion form ("Junctions + reads on target"). Significance was evaluated with respect to a FDR threshold of 5%.

**Global intron retention analysis at the gene level**. Evaluation of the relative coverage of intronic versus exonic regions at the transcriptome scale, was performed as follow. Gene annotations were downloaded from Gencode[62] (release 19) and restricted to protein-coding genes with level 1 and 2 annotations (i.e., verified or manually annotated loci), keeping only *exon* and *gene* features. For each annotated gene, quantification of *exonic reads* was obtained from the STAR alignments generated above by calculating the number of reads comprised entirely and without ambiguity within an exon of the corresponding gene using HTseq count in "intersection-strict" mode on *exon* features[63]. Quantification of *genic reads* (exonic + intronic) was obtained similarly using HTseq count on the *gene* features. Due to overlaps between genes in the genome (only reads unambiguously assigned to a single feature are counted), for a few genes, the *genic read count* was

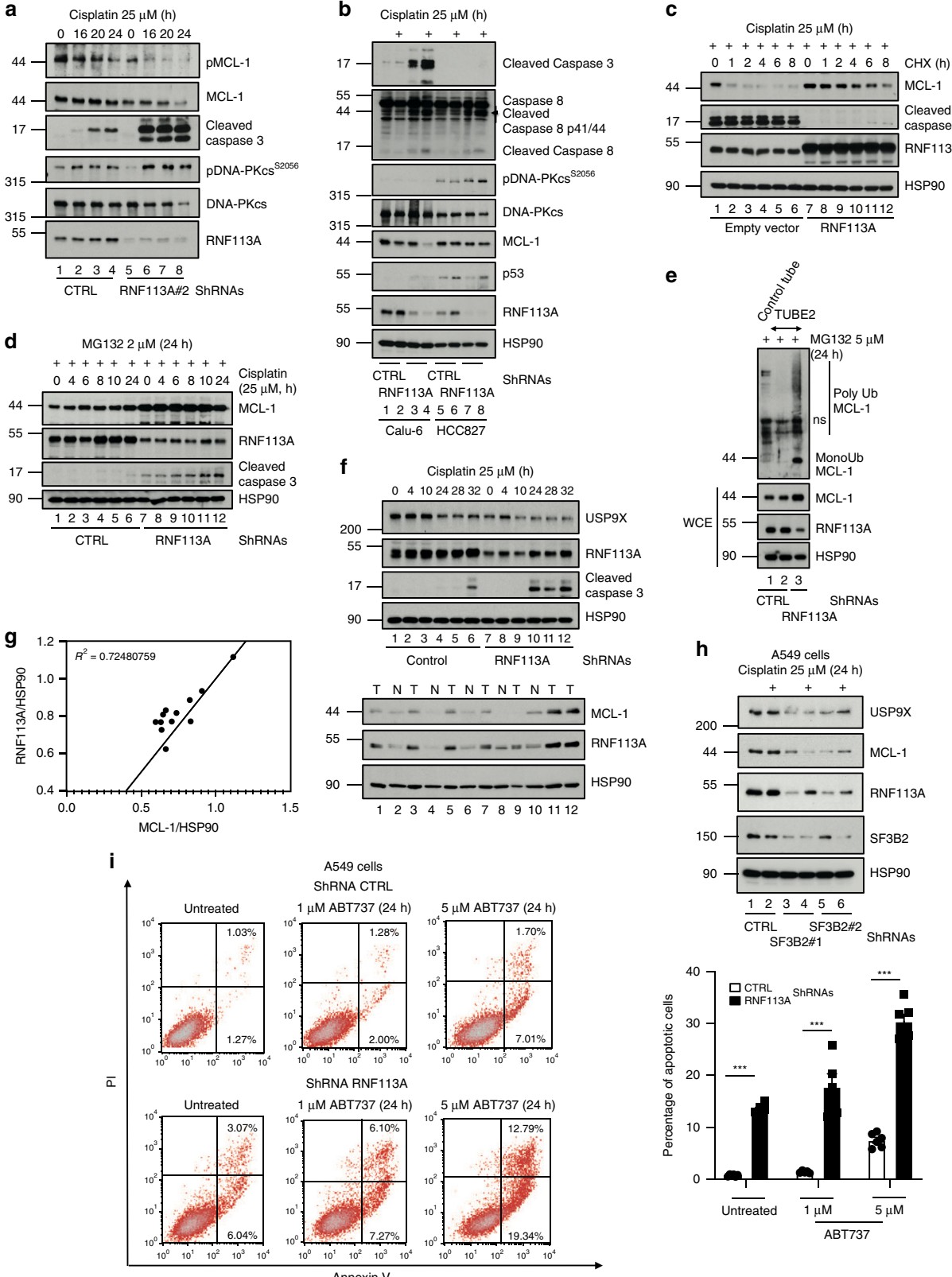

lower than the *exonic* read count. These genes (~5%) were discarded from the analyses although keeping them led to similar results (results not shown). Only genes for which at least one read was observed in every sample were included in the analyses. Given that all annotated exons were taken into consideration for the exonic read counts, the definition of exonic regions was the most comprehensive possible. The difference between the genic and the exonic read count provides thus for each gene a conservative estimate of the number of reads mapping (at least partly) to its intronic regions.

**Downstream analyses and plots**. Downstream analyses and plots were generated using the R statistical package (R Development Core Team. R Foundation for Statistical Computing, Vienna, Austria, 2011).

**Fig. 9 RNF113A stabilizes MCL-1 protein levels. a**, **b** RNF113A controls MCL-1 expression in Cisplatin-treated cells. Control or RNF113A-depleted A549 (**a**), Calu-6 or HCC827 (**b**) cells were treated or not with Cisplatin and WB analyses were done. **c** RNF113A overexpression enhances MCL-1 protein half-life in Cisplatin-treated cells. Control or RNF113A-overexpressing cells were stimulated with Cisplatin and subsequently left untreated or stimulated with Cycloheximide (CHX) (50 μg/ml). WB analyses are illustrated. **d** RNF113A limits MCL-1 degradation by the proteasome upon stimulation by Cisplatin. Control or RNF113A-deficient A549 cells were simultaneously treated with MG132 (2 μM) and Cisplatin (25 μM). WB analyses are illustrated. **e** RNF113A deficiency enhances MCL-1 polyubiquitination. Protein extracts from control or RNF113A-depleted A549 cells treated with MG132 were incubated with TUBE 2 agarose beads to trap polyubiquitinated proteins (top panel) followed by anti-MCL-1 WB analyses. Crude cell extracts were also subjected to anti-MCL-1, -RNF113A, and -HSP90 WB analyses (bottom panels). **f** USP9X levels are decreased upon RNF113A deficiency in lung cancer cells. Control and RNF113A-depleted cells were untreated or stimulated with Cisplatin. WB analyses are illustrated. **g** RNF113A and MCL-1 protein levels are positively correlated in clinical cases of lung cancer. Cell extracts from clinical cases of lung cancer (T) and normal adjacent tissues (N) were subjected to WB analyses (right panel). Expression values (relative OD, as indicated) were plotted and the R-squared was calculated (left panel). **h** SF3B2 deficiency impairs RNF113A, USP9X and MCL-1 levels in lung cancer cells. Control or SF3B2-depleted A549 cells were treated or not with Cisplatin. WB analyses are shown. **i** RNF113A deficiency sensitizes lung cancer cells to BCL-2 inhibition. Control or RNF113A-depleted A549 cells were untreated or incubated with ABT737 for 24 h. FACS analyses to quantify the percentage of cells undergoing early or late apoptosis are illustrated. On the right, FACS data from two independent experiments are illustrated in the histogram (Student t-test, ***p < 0.001).

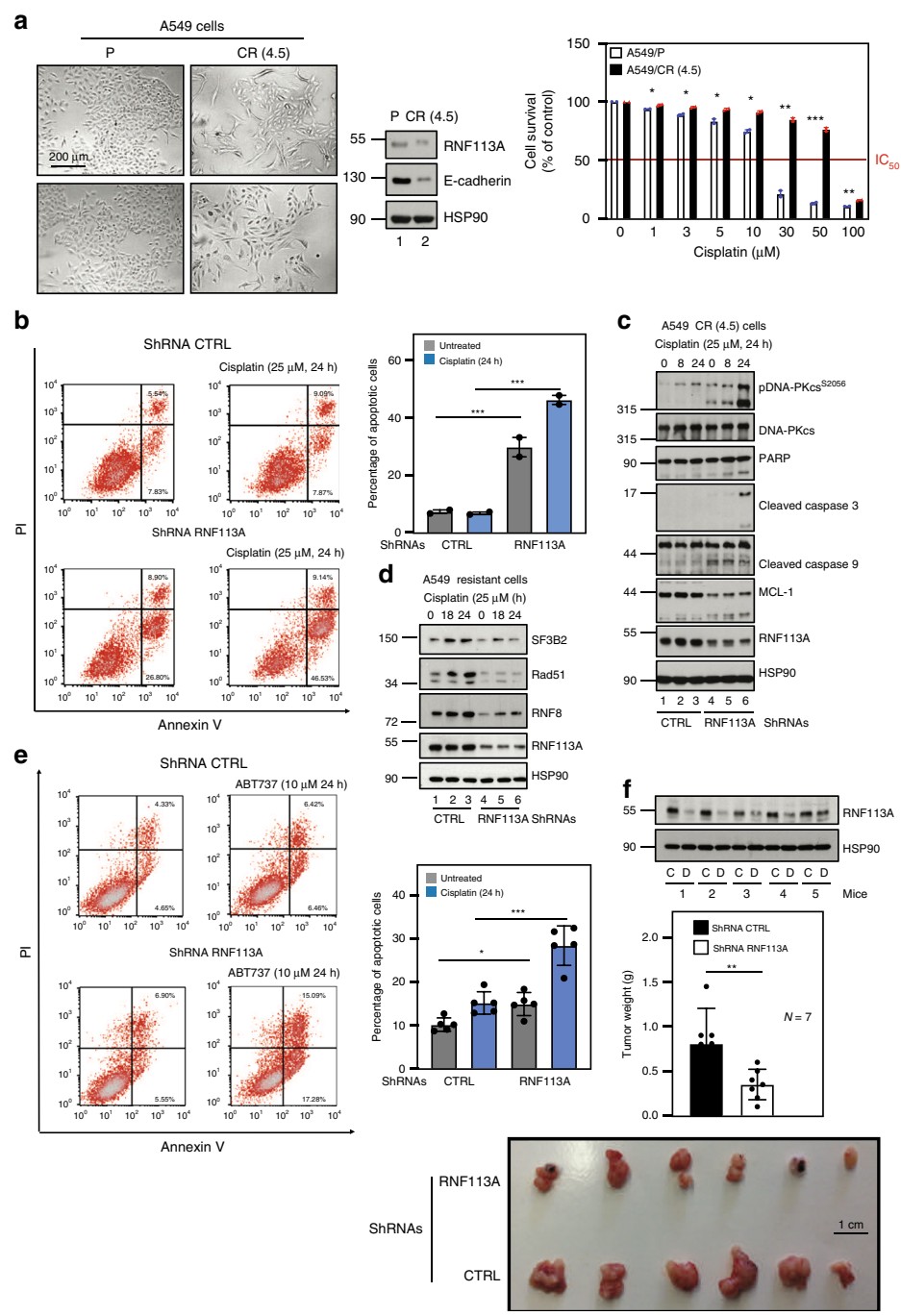

**Fig. 10 RNF113A promotes resistance to chemotherapy. a** Generation of Cisplatin-resistant lung cancer A549 cells (CR (4.5)). Pictures of both parental and resistant cells and WB analyses are shown. On the right, sensitivity of parental and resistant cells to 72 hours of treatment with Cisplatin. The percentage of viable cells in untreated parental or resistant cells was set to 100%. Data from two independent experiments in triplicates are shown (mean ± SD, ***$p < 0.001$, **$p < 0.01$, *$p < 0.05$, Student $t$-test). **b** RNF113A deficiency in CR (4.5) cells enhances cell death upon Cisplatin stimulation. On the left, cell survival upon Cisplatin treatment in resistant control and RNF113A-depleted A549 cells was assessed by FACS. The percentage of cells in early or late apoptosis is mentioned. On the right, FACS data from two independent experiments are illustrated in the histogram (Student $t$-test, ***$p < 0.001$). **c** RNF113A deficiency in Cisplatin-resistant lung cancer cells enhances DNA-PKcs phosphorylation upon DNA damage. Control or RNF113A-depleted Cisplatin-resistant A549 cells were treated or not with Cisplatin and WB analyses are shown. **d** RNF113A promotes SF3B2, Rad51 and RNF8 expression. Control or RNF113A-depleted Cisplatin-resistant A549 cells were untreated or stimulated with Cisplatin and WB analyses are shown. **e** RNF113A deficiency sensitizes Cisplatin-resistant lung cancer cells to BCL-2 inhibition. Control or RNF113A-depleted Cisplatin-resistant A549 cells were untreated or incubated with ABT737. FACS analyses were done to quantify cells undergoing early or late apoptosis. On the right, FACS data from two independent experiments are illustrated in the histogram (Student $t$-test, ***$p < 0.001$). **f** RNF113A deficiency sensitizes resistant lung cancer cells to Cisplatin-dependent cell death. Control or RNF113A-depleted Cisplatin-resistant A549 cells were transplanted into immunodeficient mice. Tumors were grown up to 0.1–0.2 mm³ and mice were treated with Cisplatin (1 mg/kg) six times every 3 days. Seven mice were used per experimental conditions. WB analyses were done with tumors generated from control (« C ») and RNF113A-depleted (« D ») Cisplatin-resistant cells. At the bottom, illustration of the size of tumors obtained after Cisplatin administration and quantification of tumor weights after Cisplatin administration (two experiments, Student $t$-test, **$p < 0.01$).

**Tumor growth in vivo**. Five millions of control or RNF113A-depleted Cisplatin-resistant A549 cells were transplanted into immunodeficient NOD/SCID 8 weeks old mice. Tumors were grown up to 0.1–0.2 mm³ and mice were then treated with Cisplatin (1 mg/kg) six times every 3 days. Seven mice were used per experimental conditions. No randomization of mice was used. Mice analyzed were litter mates and age-matched whenever possible. No blinding was done during the experiment as mice were injected with control or RNF113A-depleted cells in each flanck.

**Study approval**. The ethical comity of the University of Liège (CHU, Sart-Tilman) approved all experiments carried out with mice. A written informed consent was received from participants at the University Hospital of Cologne prior to inclusion in the study.

**Reporting summary**. Further information on research design is available in the Nature Research Reporting Summary linked to this article.

## Data availability

Raw data and results of the RNAseq analysis are available on the NCBI GEO website under the GSE133029 accession number. Source data for all figures are provided as a Source Data file. All data is available from the authors upon reasonable request.

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

## Acknowledgements

The authors are grateful to Dr. Gaëlle Legube (University of Toulouse, Toulouse, France) for the gift of DIvA U2-OS cells. We also thank the GIGA Imaging and Flow Cytometry Facility as well as the GIGA Genomics Platform for RNA-Sequencing analyses. This study was supported by Grants from the Belgian National Funds for Scientific Research (FNRS), from the Concerted Research Action Program (UBICOREAR) and Special Research Funds (FSR) at the University of Liege, the Belgian foundation against Cancer (FAF-F/2016/794), as well as from the Walloon Excellence in Life Sciences and Bio-technology (WELBIO-CR-2015A-02). We are also grateful to the "Fonds Leon Fredericq" and the "Centre Anticancéreux" of the CHU Liege for their financial support. A. Chariot and P. Close are Research Director and Research Associate at the FNRS, respectively.

## Author contributions

K.S., Z.J., L.T., X.X., H.Q.D., S.K., A.V., and A.F. conducted experiments; Y.H. provided reagents and suggested experiments, B.C., A.M., K.S., F.R., J.C.M., P.C., and R.B. acquired and analyzed data; A.C. and K.S. designed research studies; A.C. wrote the manuscript.

## Competing interests

The authors declare no competing interests.
