## [Peer Review File · Nature Communications]

Reviewers' comments:

Reviewer #1 (Remarks to the Author):

In this manuscript Shostak and colleague show the upregulation of the RING finger protein 113A (RNF113A), a E3 ligase, in lung cancer tissue and in lung cancer cells exposed to DNA damage agents. RNF113A expression prevents DNA-damage response activation in lung cancer cells and promotes DNA damage repair, thus preventing cancer cells from genotoxic stress induced-cell death. In their discussion authors claim: "We show here that RNF113A, whose expression increases through a C/EBP β dependent pathway upon DNA damage, promotes cell survival as a spliceosome subunit"; however none of this statement is fully supported by authors finding: 1- Dependency of RNF113A expression on C/EBP β dependent pathway should be probed, by direct testing of its binding to RNF113A promoter. Moreover, the rationale for choosing to investigate C/EBP β is unclear, as this is not the only, nor the main, transcription factor involved in DDR. Lastly, the increase in RNF113A protein expression appears to precede up-regulation of its mRNA in A549 cells (Fig.1D and 1G). Thus, regulation of RNF113A expression may also depend on post-transcriptional mechanisms.

2- Figure 3: the authors conclude: "Therefore, DNA damage and cell apoptosis triggered by Cisplatin correlates with the shuttling of RNF113A to the cytoplasm in order to potentially limit cell death." On what is based this assumption? They show that RNF113A is recruited to foci of DNA together with DNAPK, how would a cytoplasmic localization of the protein limit cell death?

3- Data shown by the authors do not unambiguously correlate the pro-survival activity of RNF113A to its splicing activity, but they show RNF113a as a multifunctional protein regulating by different mechanism cell survival.

Its E3-ligase activity is indeed important in the control of MCL1 protein stability and, although no clear-cut insights about the underlying regulatory mechanisms are provided, RNF113a regulates NHEJ factors recruitment to DNA damage sites.

Also the reduction of SF3B2 expression levels elicited by RNF113a, which in turn enhances R-loop and DNA damage formation, is not clearly ascribed to an impairment of the processing of its transcript or to a post-transcriptional control, as shown for MCL-1.

A direct link between the pro-survival activity of RNF113a and its splicing activity might be provided by rescue experiment with ASO modulating its splicing target of vectors overexpressing its regulated splice variants.

Overall the manuscript shows plenty of data, but the work is poorly organized and frequently misleads the reader. Figures should be re-organized, adding proper graph labeling, to facilitate readers understanding of experiments and results.

Specific points:

Figure 1, Panel A, how was the percentage of tumor cells within the sample estimated? No-description of this methodology is available neither in the material and methods sections, nor in the figure legend or main text.

On which kind of analysis of RNF113A expression levels in lung cancer tissue (IHC, WB, RT-PCR) was the Kaplan Meier Analysis performed? Please clarify.

To ascribe to C/EBP β activity the upregulation of RNF113A following cisplatin treatment, its binding to RNF113A promoter should be shown. Missed-upregulation of RNF113A in C/EBP β knocked-out cells might indeed be an indirect effect of an impaired DNA-damage response activation, unable to elicit by other possible mechanisms RNF113A upregulation.

Figure 2, Panel A. Upregulation of RNF113A following cisplatin exposure is not appreciable in the western blot shown. Panel B. Which parameters were analyzed in the FACS analysis to detect cell-death of cisplatin-treated cells? Please specify both in the figure and in the figure legend. It is also unclear why the authors address only DNAPK as DDR kinase (Fig.2), and not also ATM and ATR. Is

RNF113A expression specifically required for this pathway or is this the only one that they have investigated? In this paragraph, the authors claim that they have “established a role of RNF113A in DNA repair”. However, a role for this protein in DNA repair in response to DNA alkylating agents was already demonstrated (Brickner et al., *Nature*, 2017).

Figure 4 Panels E,F. To properly judge whether RNF113A depletion impairs the interaction between SF3B1 and SF3B2, the amount of immunoprecipitated SF3B1 protein should be shown. How does RNF113a depletion down-regulate SF3B2 expression? Is it a target of its E3-ligase activity or does RNF113a modulate SF3B2 mRNA processing, as shown for Rad51?

Figure 5, Panel E. it is not clear which kind of different transcripts the three graph represent neither if the difference shown by the bar graph are statistically significant.

Panel F. This PCR analysis doesn't demonstrate that RNF113a depletion promotes an alternate intron 1 in RNF166. PCR analysis also for a constitutive and non-regulated mRNA region of RNF113A should be shown. Quantification by qPCR analyses or at least densitometric analyses of agarose gel should corroborate the data.

Figure 6. To which of the two alternative transcripts of SAT1 and NUPR1 genes correspond the different columns of graphs in panel A and F? In Graph C and H it should be indicated which transcript of the SAT1 and NUPR1 genes is analyzed, not only in the figure legend.

Figure 8. although MCL-1 stability appears to rely on RNF113A, it is unclear whether this requires assembly of RNF113A in the spliceosome complex, as the title of the article suggests.

Reviewer #2 (Remarks to the Author):

This manuscript describes a series of experiments to evaluate the molecular function of the still poorly characterized ubiquitin ligase RNF113A. Using a rather complete series of approaches the authors convincingly highlight a role for RNF113 in the control of pre-mRNA splicing upon DNA damage induced by cisplatin.

Interestingly, the authors further demonstrate that silencing of RNF113A could be instrumental for the treatment of cisplatin-resistant tumours. The experimentation is nicely multi-disciplinary and in general technically sound.

However, the manuscript is quite dense at places and there are issues that preclude a recommendation to publish at *Nat Comm*. without a major revision. There may be a publishable story here, but the authors need to do a much better job of organizing and presenting their data and conclusions.

1. The Introduction needs a major revision. It lacks a proper discussion of the mechanism of damage induced by Cisplatin. Instead of discussing the implication of NER, the major repair pathway of the ISCs induced by cisplatin the authors discuss DSB repair pathways that may be only a secondary event caused by the arrest of the replication fork at ISCs. The fact that the authors did not consider an implication of NER is really surprising considering that RNF113A is a trichothiodystrophy linked gene, and trichothiodystrophy is a well-known genetic disease linked to NER deficiency.

For this reason, I would suggest to assess in the experiments shown in Figures 1 and 2 the activation of additional components of the DNA damage response such as ATM and ATR that are activated by NER components, in addition to DNPK.

2. Fig 1g-h. The authors should comment on the different effects of camptothecin (a Top1 inhibitor) and etoposide (a Top2 inhibitor) on RNF113A expression. This suggest a difference in the repair pathway that is activated upon DNA damage.

3. According to the authors RNF113A colocalizes with phosphoDNAPK but not with gH2AX foci, a well-established marker of DNA damage. Why is that? Why did the authors use RNase A extraction?

4. To support a role for RNF113A in the recruitment of DSB repair factors at site of damage the

authors should perform ChIP experiments. Several different methods and cell lines exist that allow monitoring of protein recruitment at specific genomic loci at which DSB are induced by endonuclease cleavage.

5. Figure 3d right panel. Apparently the labelling of the lanes is inverted.

6. Figure 3g. I do not understand why only one cell is positive for H2AX considering that "Immunofluorescence analyses were conducted to define the subcellular localization of RNF113A in Cisplatin-treated A549 cells." As the authors state in the figure legend.

7. In general, the IF images are quite poor. Specifically, I have concerns about the quantification of R-loops performed in Figures 4. Importantly, I am concerned because of the lack of nucleolar staining in control cells. This staining has been reported by several authors and is usually quite resistant to RNase H1 digestion (for expl. see Sollier, Mol Cell, 2014). I would suggest to more reliably quantify nucleoplasmic R-loop levels as described for ex. in Hodroji et al, EMBOJ 2017, i.e. using the DAPI signal to create a mask of the nucleus and then to determine the nuclear S9.6 signal intensity by subtracting the nucleolar signal (nucleolin staining) and analyzing the remaining S9.6 signal. Quantification should be performed for 100-200 cells per experiment.

8. P. 7 lines 147-148 "Accordingly, RNF113A deficiency enhanced the number of both phospho-H2AX (pH2AX) and phospho-DNA-PKcs positive cells".

As for R-loops quantification, the way the authors performed this experiment is puzzling. To quantify the total intensity in the whole nucleus then z-stacks should be analysed and not just one arbitrary plane. More appropriately, the authors should quantify the number of foci per cell in untreated and treated cells for at least 150 cells per experimental group.

9. 148 (pDNA-PKcs) positive BZR-T33 cells, sAlso the way activation of the DDR was assessed by IF puzzles me.

10. All the experiments assessing splicing of specific transcripts lack quantifications (Figs. 5f, 6b, 6g, 7a).

11. The bioinformatics analysis performed on the RNAseq dataset is poor, in particular for what the splicing changes are concerned. How does RNF113A silencing affects splicing? Does it promote intron retention, inclusion/skipping of alternative exons?

12. To further support the role of RNF113A on NOX1 activity the authors could complement .NOX1 in RNF113A deficient cells (Fig. 7d)

13. The methods section is incomplete. Several experimental approaches are not described. How was the phosphorylation of H2Ax quantified by IF? How were R-loops measured?

Minor points

Sometimes the rational for specific experiments is not well explained, so that the paper is difficult to read for non-specialized readers. For example, it would not harm to explain the link between ERK1/2 and MEK1/2 (page 6, line 117-118)

Spelling mistakes throughout: for ex. p19, line 435 the camptothecin concentrations are too high (mM or microM?) or p. 6, line 121: "RNF113A and C/EBP expression upon ERK1/2 inhibition (was) very highly similar,"

Reviewer #3 (Remarks to the Author):

In the present manuscript the authors analyze the impact of RNF113A, a E3 ligase and a spliceosome subunit, in gene expression and DNA damage response (DDR) upon cisplatin treatment. The subject is of interest but data quality and presentation is questionable. In general terms the authors want to conclude that RNF113A is a splicing factor relevant for the DDR, but in my opinion they fail to properly conclude this although some of the data might be promising. Therefore, at least in its current form, the work is not suitable for publication in nature communications.

Major points:

- If RNF113A controls the pool of NHEJ factors recruited to damaged DNA, then the authors should work with cells arrested in G1, so as to avoid replication issues, and use a true DSB inductor, not cisplatin, to be able to say that RNF113A controls the pool of the recruited NHEJ factors. This applies also to other relevant results such as the proposed differential binding of RNF113A to mRNAs after damage (figure 4) or the effect on cell survival upon knock down / overexpression of RNF113A (figure 2).
- When there is no over-expression is not clear at all that cisplatin reduces the amount of RNF113A to oligodT bounded RNAs (Fig 4a, lines 1 and 3) since it looks like there is more binding (lines 1 vs 3) but also the input is affected (lines 5 vs 7). Conclusions about differential binding in normal or cisplatin treated cells should be done in cells with no RNF113A overexpression. Also, proper quantitation of nuclear and cytoplasmic RNF113A in control and cisplatin treated no over-expressing cells should be done.
- Focusing on splicing and alternative splicing, the authors should just present the percentage of alternative exon cassette inclusion (and maybe another forms of AS that are less common) and/or unspliced transcripts upon cisplatin and RNF113A knock down in the whole transcriptome. Then they should specifically analyze the subset of relevant genes i.e. death related genes. The way the authors present the data is rather confusing and this is not just in figure 5, but throughout the hole text (see below).
- The authors say that RNF113A is more expressed upon cisplatin but also more localized in the cytoplasmic fraction (while splicing takes place in the nucleus). They also say that upon cisplatin, RNF113A is less bound to pre-mRNAs but, at the same time, RNF113A is necessary to properly express key mRNAs isoforms like, for instance, in the Nupr1 gene. To this referee this is not clear at all.

Minor points:

The manuscript have many errors and omissions that makes data interpretation difficult.

Bellow some examples,

Fig 1 a, what is the ratio RNF113A / actin?

Fig 1b, what is the purpose of this figure? What is the localization of RNF113A in normal cells? Is there any function of RNF113A in cytoplasm?

Fig 1 d/e, now hsp90 is the loading control but in fig 1a was actin. The authors should use the same factor as a loading control throughout the manuscript. Moreover, overexposed western blots should be replaced.

Fig 1 d/e, cisplatin treatment enhanced RNF113A in cancer cells lines, and in non-transformed cells? Is there any point in stressing "in cancer cells" or it's a general phenomenon after cisplatin treatment? What about Ku80 wb? Is not commented anywhere in the text.

Fig 1 i/j Unless there is a point in analyzing ERK1/2 dependency in RNF113A expression in different cells lines the figure should be removed.

Fig 2a, the legend states that the western was carried out 20 hours after cisplatin treatment but the figure shows a time course.

Fig 2b, only one experiment should be shown, otherwise the figure is misleading.

Fig 2d, the authors say that "RNF113A deficiency increases the inhibitory effect of Cisplatin on the clonogenicity of A549 cells". Is there any interaction between cisplatin and shRNA treatment? Can the authors really say that the cisplatin effect is increased upon RNF113 deficiency? It is clear that there are less cells in shRNA treated cells with no cisplatin.

Fig 3d, In the chromatin fraction the RNF113A levels are wrong or the legend is wrong.

Fig 4a/b, RNaseA is written in different ways in figures 4a and 4b.

Fig 5b, Figure legends and text doesn't match. Control bars are #2 and #4???? Or is the other way around?

Fig 5f, this is wrong, there is no alternative intron (as stated in the text, page 11 line 254) nor alternative exon (as written in the figure) but alternative promoters as is always the case when exon 1 is different between two transcripts sharing the rest of the mRNA sequences. The end point PCR showed at the bottom right of the figure shows no quantitation and therefore is not clear if RNAseq results are validated or not. Of course, the authors should use a qPCR approach to be able to talk about different quantities of RNF116 transcript upon shRNA treatment or cisplatin treatment. Moreover, and once again, the legend of the figure is wrong since it is shown "cisplatin cisplatin" on top of the gel where the correct sentence is probably "control cisplatin".

Fig 6b, once again the figure is hard to understand. IT is not clear what is the treatment of each sample with respect to untreated or cisplatin treated and shRNAs. A ratio largo/short isoforms should be presented.

Fig 6f, why the authors are show 8 bars in the graph? What is the difference between the first and the last 4 bars? I see no explanation in the text or in the figure legend. This is happening quite often in this work, understanding figures is not an easy task.

Fig 8a, there is no loading control? The how can the authors and readers properly conclude that MCL is degraded in the absence of RNF113A?

Reviewers' comments:

Reviewer #1 (Remarks to the Author):

In this manuscript Shostak and colleague show the upregulation of the RING finger protein 113A (RNF113A), a E3 ligase, in lung cancer tissue and in lung cancer cells exposed to DNA damage agents. RNF113A expression prevents DNA-damage response activation in lung cancer cells and promotes DNA damage repair, thus preventing cancer cells from genotoxic stress induced-cell death. In their discussion authors claim: "We show here that RNF113A, whose expression increases through a C/EBP β dependent pathway upon DNA damage, promotes cell survival as a spliceosome subunit"; however none of this statement is fully supported by authors finding:

Our answer:

We will provide point-by-point answers to Reviewer 1' comments here after. We think that our conclusions are supported by our findings. Indeed, regarding the link of RNF113A with the spliceosome, we believe that a major finding is the fact that SF3B2 deficiency mimics RNF113A deficiency (enhanced cell death upon DNA damage, destabilized MCL-1 levels, defective induction of NUPR1 expression by Cisplatin for example). Although we agree that it is an indirect evidence, we do not think that these observations would have been seen if RNF113A was not acting as a spliceosome subunit. We will also provide a variety of additional experiments to further support our conclusions (see here after).

1- Dependency of RNF113A expression on C/EBP β dependent pathway should be probed, by direct testing of its binding to RNF113A promoter.

Our answer:

We identified multiple C/EBP β binding sites on the RNF113A promoter (see our new Figure 1j) and we also carried out ChIP assays to show that C/EBP β is recruited on this promoter, both in unstimulated and in Cisplatin-treated A549 cells. This new result, combined with the demonstration that C/EBP β deficiency impairs RNF113A expression (Figure 1i), define C/EBP β as a protein that controls RNF113A expression.

Moreover, the rationale for choosing to investigate C/EBP β is unclear, as this is not the only, nor the main, transcription factor involved in DDR. Lastly, the increase in RNF113A protein expression appears to precede up-regulation of its mRNA in A549 cells (Fig.1D and 1G). Thus, regulation of RNF113A expression may also depend on post-transcriptional mechanisms.

Our answer:

We agree that C/EBP β is probably not the only transcription factor that controls RNF113A expression as multiple binding sites for other candidates were actually found in our analyses (AP1, ...etc..) but we did not want to concentrate too much on this issue as we wanted to rather concentrate on the still poorly understood biological roles of RNF113A. Regarding the mechanisms by which RNF113A expression is induced by Cisplatin at early time points, we do not feel 100% comfortable to compare the kinetics of RNF113A induction using two distinct experimental approaches, namely western blots and real-time PCR analyses. We agree with Reviewer 1 that other mechanisms, including protein stabilization, may occur upon Cisplatin stimulation but we currently do not have any data supporting this claim so we decided not to mention this possibility at this stage. It is also fair that we are already reaching the limit of 10 figures requested by « Nature Communications » in our revised manuscript.

2- *Figure 3: the authors conclude:” Therefore, DNA damage and cell apoptosis triggered by Cisplatin correlates with the shuttling of RNF113A to the cytoplasm in order to potentially limit cell death.” On what is based this assumption? They show that RNF113A is recruited to foci of DNA together with DNAPK, how would a cytoplasmic localization of the protein limit cell death?*

Our answer:

We rephrased our conclusion in the revised manuscript to avoid any confusion by removing the sentence « in order to potentially limit cell death ».

3- *Data shown by the authors do not unambiguously correlate the pro-survival activity of RNF113a to its splicing activity, but they show RNF113a as a multifunctional protein regulating by different mechanism cell survival. Its E3-ligase activity is indeed important in the control of MCL1 protein stability and, although no clear-cut insights about the underlying regulatory mechanisms are provided, RNF113a regulates NHEJ factors recruitment to DNA damage sites. Also the reduction of SF3B2 expression levels elicited by RNF113a, which in turn enhances R-loop and DNA damage formation, is not clearly ascribed to an impairment of the processing of its transcript or to a post-transcriptional control, as shown for MCL-1.*

Our answer:

As SF3B2 deficiency perfectly mimics the phenotype obtained upon RNF113A deficiency, we believe that we can conclude that RNF113A acts as a splicing factor that promotes cell survival upon DNA damage. We actually believe that RNF113A expression is critical for the integrity of some subunits of the spliceosome as SF3B2 expression is indeed decreased at the protein level in A549 cells upon RNF113A deficiency (Figure 4d) and the binding between SF3B1 and SF3B2 is also impaired upon RNF113A deficiency in BZR-T33 cells (Figure 4f). Regarding the molecular mechanisms by which SF3B2 expression

is decreased upon RNF113A deficiency, we now show here after that SF3B2 transcription and splicing are unaltered upon RNF113A deficiency, which strongly suggests that this regulation occurs at the protein level. Note that we decided not to show this new data in the revised manuscript because of a lack of space.

C

Figure legend and results: SF3B2 is not a splicing target of RNF113A. **a.** Illustration of an IGV histogram. Very minor reads occurs between exons 5 and 6 and exons 19 and 20 in our analysed samples (ShRNA RNF113A and ShRNA RNF113A treated with Cisplatin, when compared with ShRNA Control, see the red boxes). **b.** and **c.** RNF113A deficiency does not impact on SF3B2 mRNA levels and splicing. As there is currently no data available in the literature regarding any potential SF3B2 variants that result from alternative splicing, we checked Ensembl data (www.ensembl.org/index.html) and found some variants (with Transcript Support Level 1 or 2- supported by mRNA or EST reads) with retained introns as well as with excluded exon 5 (in red in the schematic representation). The mRNA which includes exon 5 encodes a 895 amino acids protein while the transcript lacking this exon is expected to encode a 878 amino acids polypeptide. To detect potential changes in the levels and/or splicing of the SF3B2 transcript upon RNF113A deficiency, we designed primers within exons 10 and 13 and first concluded that RNF113A deficiency did not have any impact on

mRNA levels of SF3B2 (see Histogram illustrated in b.). Using primers within exons 5 and 7 (cf schematic representation of the SF3B2 transcript), we carried out some RT-PCRs (c) and concluded that RNF113A deficiency did not have any impact on SF3B2 splicing, both in control and Cisplatin-treated cells.

A direct link between the pro-survival activity of RNF113a and its splicing activity might be provided by rescue experiment with ASO modulating its splicing target of vectors overexpressing its regulated splice variants.

Our answer:

We agree with Reviewer 1 that this strategy can be useful to demonstrate that cell death seen upon RNF113A deficiency is indeed due to splicing defects. Note however that we know that RNF113A has multiple targets. As a result, it is unlikely that the restoration of an unaltered splicing for one single candidate would restore cell survival. Doing ASOs for all candidates at the same time is unfortunately not feasible. Nevertheless, we selected Noxa1 as one candidate whose splicing changes upon RNF113A deficiency. We succeeded in partially decreasing the alternative spliced variant but unfortunately, we could not completely restore initial levels of the full-length protein seen in control cells. Our results are illustrated below: FACS analysis of cell death (Annexin/PI) with partial rescue and western blots showing changes in the ratio between the alternatively spliced and the full transcript using several ASO constructs. Note that Lipofectamine 3000 used in our protocol is quite toxic, especially for cells previously subjected to lentiviral infections.

Figure legend: Consequences of Noxa1 splicing on cell survival upon DNA damage. Control or RNF113A-depleted A549 cells were plated in 12-wells dishes and transfected with 10mM of ASO sequences using Lipofectamine 3000, according to the manufacturer's protocol. Cells were harvested after 72 hours and extracts were subjected to western blot analyses. Cells were also plated in 24-wells dishes, transfected in triplicates for each experimental condition, treated or not with Cisplatin (25 μ M) for 20 hours and subjected to FACS analyses (AnnexinV/PI staining) to assess cell death.

Overall the manuscript shows plenty of data, but the work is poorly organized and frequently misleads the reader. Figures should be re-organized, adding proper graph labeling, to facilitate readers understanding of experiments and results.

Our answer:

We went through the entire manuscript to facilitate readers understanding.

Specific points:

Figure 1, Panel A, how was the percentage of tumor cells within the sample estimated? No-description of this methodology is available neither in the material and methods sections, nor in the figure legend or main text. On which kind of analysis of RNF113A expression levels in lung cancer tissue (IHC, WB, RT-PCR) was the Kaplan Meier Analysis performed? Please clarify.

Our answer:

We now provide some precise explanations on how the percentage of tumor cells within each sample was estimated (see Figure legend for details). This approach is routinely used in the Laboratory of Pathology (University Hospital of Cologne, Germany). Regarding the Kaplan Meier analysis established based on RNF113A expression in lung cancer tissues, this was established by RT-PCR. This information was added in the Figure legend as well.

To ascribe to C/EBP β activity the upregulation of RNF113A following cisplatin treatment, its binding to RNF113A promoter should be shown. Missed-upregulation of RNF113A in C/EBP β knocked-out cells might indeed be an indirect effect of an impaired DNA-damage response activation, unable to elicit by other possible mechanisms RNF113A upregulation.

Our answer:

We carried out the requested ChIP assays to show that C/EBP β is recruited on this promoter, both in unstimulated and in Cisplatin-treated A549 cells.

Figure 2, Panel A. Upregulation of RNF113A following cisplatin exposure is not appreciable in the western blot shown. Panel B. Which parameters were analyzed in the FACS analysis to detect cell-death of cisplatin-treated cells? Please specify both in the figure and in the figure legend. It is also unclear why the authors address only DNAPK as DDR kinase (Fig.2), and not also ATM and ATR. Is RNF113A expression specifically required for this pathway or is this the only one that they have investigated? In this paragraph, the authors claim that they have “established a role of RNF113A in DNA repair”. However, a role for this protein in DNA repair in response to DNA alkylating agents was already demonstrated (Brickner et al., Nature, 2017).

Our answer:

We re-made the experiment illustrated in Figure 2a and we now show that RNF113 is indeed induced by Cisplatin, as also shown in all other panels. We thank Reviewer 1 for pointing-out this issue. Regarding the parameters used to quantify cell death by FACS, we added the following sentence in the legend : “On the left, representative FACS analyses in which PI and/or Annexin V-positive cells are quantified. The percentage of cells in early (Annexin V positive and PI negative) or late apoptosis (Annexin V positive and PI positive) is mentioned”. We also added these parameters in Figures 2b and 2c. Regarding the activation of other kinases involved in DNA repair, we now show that ATR activation, as assessed through Chk1 phosphorylation, is also defective upon RNF113A deficiency (see our **new Figure 2g**). Therefore, multiple kinases are not properly regulated upon RNF113A deficiency. Brickner and colleagues indeed established a link between RNF113A and the DNA damage response. We provide additional insights on this issue by showing the profile of both DNA-PKcs and ATR upon RNF113A deficiency and we also concentrated our study on the role of RNF113A as a splicing factor and on the transcripts whose splicing is regulated by RNF113A, two issues that were not addressed in that study.

Figure 4 Panels E,F. To properly judge whether RNF113A depletion impairs the interaction between SF3B1 and SF3B2, the amount of immunoprecipitated SF3B1 protein should be shown. How does RNF113a depletion down-regulate SF3B2 expression? Is it a target of its E3-ligase activity or does RNF113a modulate SF3B2 mRNA processing, as shown for Rad51?

Our answer:

We now provide the requested panel in the revised Figure (now Figure 4e). The molecular mechanisms by which RNF113A controls SF3B2 expression is not at the transcriptional or splicing level (see our figure here before).

Figure 5, Panel E. it is not clear which kind of different transcripts the three graph represent neither if the difference shown by the bar graph are statistically significant. Panel F. This PCR analysis doesn't demonstrate that RNF113a depletion promotes an alternate intron 1 in RNF166. PCR analysis also for a constitutive and non-regulated mRNA region of RNF113A should be shown. Quantification by qPCR analyses or at least densitometric analyses of agarose gel should corroborate the data.

Our answer:

The analysis previously shown in Figure 5e has been replaced by a more specific analysis of alternative splicing using rMATS (**Figures 6a-d and S7a-b** in the revised manuscript). In rMATS, splicing events are analysed individually and by categories of events (skipped exons, mutually exclusive exons, retained intron, alternative 5' splicing site and alternative 3' splicing site). The

manuscript has been updated accordingly and sections relative to the RNAseq data analyses have been rewritten extensively. Previous Figure 5f has been removed from the manuscript because this analysis was distractive from our main message. Due to space limitation, we decided to remove our data on RNF166 as we gave priority to candidates identified through our new unbiased approaches.

Figure 6. To which of the two alternative transcripts of SAT1 and NUPR1 genes correspond the different columns of graphs in panel A and F? In Graph C and H it should be indicated which transcript of the SAT1 and NUPR1 genes is analyzed, not only in the figure legend.

Our answer:

Figure 7 (former Figure 6) panels relative to *SAT1* and *NUPR1* genes have been modified and clarified. Figure 7b now presents the results of a new analysis with rMATS assessing notably *SAT1* exon 4 inclusion levels.

Figure 8. although MCL-1 stability appears to rely on RNF113A, it is unclear whether this requires assembly of RNF113A in the spliceosome complex, as the title of the article suggests.

Our answer:

A key finding is the fact that SF3B2 deficiency also decreases MCL-1 protein levels (see Figure 9f). As SF3B2 is acting as a spliceosome subunit, this results strongly suggests that the spliceosome integrity is required for MCL-1 stability/expression. Let's note that our title is only talking about cell survival but does not specifically mention MCL-1. We feel comfortable to talk about cell survival in the title as SF3B2 deficiency perfectly mimics RNF113A deficiency.

Reviewer #2 (Remarks to the Author):

This manuscript describes a series of experiments to evaluate the molecular function of the still poorly characterized ubiquitin ligase RNF113A. Using a rather complete series of approaches the authors convincingly highlight a role for RNF113 in the control of pre-mRNA splicing upon DNA damage induced by cisplatin. Interestingly, the authors further demonstrate that silencing of RNF113A could be instrumental for the treatment of cisplatin-resistant tumours. The experimentation is nicely multi-disciplinary and in general technically sound. However, the manuscript is quite dense at places and there are issues that preclude a recommendation to publish at Nat Comm. without a major revision. There may be a publishable story here, but the authors need to do a much better job of organizing and presenting their data and conclusions.

1. The Introduction needs a major revision. It lacks a proper discussion of the mechanism of damage induced by Cisplatin. Instead of discussing the implication of NER, the major repair pathway of the ISCs induced by cisplatin the authors discuss DSB repair pathways that may be only a secondary event caused by the arrest of the replication fork at ISCs. The fact that the authors did not consider an implication of NER is really surprising considering that RNF113A is a trichothiodystrophy linked gene, and trichothiodystrophy is a well-known genetic disease linked to NER deficiency. For this reason, I would suggest to assess in the experiments shown in Figures 1 and 2 the activation of additional components of the DNA damage response such as ATM and ATR that are activated by NER components, in addition to DNAPK.

Our answer:

We modified our Introduction section by adding some words on mechanisms by which Cisplatin induces some DNA damage (see text marked in yellow). Regarding the activation of other kinases involved in DNA repair, we now show that ATR activation, as assessed through the phosphorylation of Chk1, the bona fide substrate of this kinase, is also defective upon RNF113A deficiency (see our new Figure 2g). Therefore, multiple kinases are not properly regulated upon RNF113A deficiency. Note that we did not observe any dramatic effect of RNF113A deficiency on ATM activation upon DNA damage, which is the reason why we do not talk about it in our study.

2. Fig 1g-h. The authors should comment on the different effects of camptothecin (a Top1 inhibitor) and etoposide (a Top2 inhibitor) on RNF113A expression. This suggest a difference in the repair pathway that is activated upon DNA damage.

Our answer:

Indeed, it is well established that Top1 and Top2 inhibitors differently activate some repair pathways due to different topoisomerase activity (Pommier et al, Chem Biol, 2010; Pommier et al, Chem Biol, 2013; DE Deweese, Biochem Mol Biol Educ 2009; JL Nitiss, Nat Rev Cancer 2009; Y. Maede et al,

Molecular Cancer Therapeutics, 2014). Their inhibitors show distinct effects on gene expression in different cell lines (and genomics context), as shown earlier (CT Huang et al, iScience, 2018). The induction of RNF113A mRNA levels by Camptothecin (a Topoisomerase 1 inhibitor), but not Etoposide (a Topoisomerase 2 inhibitor) could be explained by the fact that Top1 occupancy is linked to active transcription and control of histone density while Top2 binding does not correlate with transcription (M. Durand-Dubief. et al, EMBO J, 2010). Inhibition of Top1 may result in the down regulation of factors suppressing RNF113A mRNA expression or changes in the promoter region of the *rnf113a* gene itself. As these issues are currently speculative and need further investigation, we prefer not to talk about them in our revised version.

3. According to the authors RNF113A colocalizes with phosphoDNAPK but not with *gH2AX* foci, a well-established marker of DNA damage. Why is that? Why did the authors use RNase A extraction?

Our answer:

It was shown, using ultra high resolution microscopy that foci of pH_2AX and pDNA-PKcs as well as Ku80/70 did not totally co-localize (D. Sisario et al, FASEB Journal, 2018 and S. Britton et al, Journal of Cell Biology, 2013). S. Britton and colleagues showed in that paper that the use of RNase A was required in immunofluorescence analyses of NHEJ factors (pDNA-PKcs, Ku70, Ku 80 and Rad51) in order to remove some background staining since some of these factors may bind RNA (Drouet et al, The Journal of Biological Chemistry, 2005). Since RNF113A is also a RNA binding protein, we decided to use pre-extraction with the CSK buffer (to release soluble proteins) and to carry out some RNase A treatment to see its possible role in DNA repair as well as its potential co-localization with these factors.

4. To support a role for RNF113A in the recruitment of DSB repair factors at site of damage the authors should perform ChIP experiments. Several different methods and cell lines exist that allow monitoring of protein recruitment at specific genomic loci at which DSB are induced by endonuclease cleavage.

Our answer:

We carried out the requested experiment using the DivA U2-OS cell line, which stably expresses the restrictase AsiSI under *ER* promoter. This mammalian cell line generates several randomly distributed and sequence-specific DSBs (Lacovoni JS and colleagues, EMBO Journal, 2010, 29, 1446-1457). As expected, treatment of this cell line with 4-hydroxy tamoxifen (4OHT) generated DSBs since multiple pH_2AX^+ cells were detected by immunofluorescence (see our new Figure S5). We therefore generated control and RNF113A-depleted cells (Figure S5). ChIP assays were conducted to assess the presence of pH_2AX on AsiSI sites in both control and RNF113A-depleted cells using appropriate primers. Interestingly, the presence of pH_2AX

on H₂AX-associated AsiSI sites using primers 183, 906, 307 and 221 was defective upon RNF113A deficiency (Figure 3f). As negative controls, we also conducted these experiments using primers 811 and 903, which are not H₂AX-associated AsiSI sites (Figure 3f).

5. Figure 3d right panel. Apparently the labelling of the lanes is inverted.

Our answer:

The labelling of the lanes is correct. We were indeed surprised to notice that RNF113A deficiency did not have much impact on protein levels of RNF113A associated to the chromatin. Our depletion of RNF113A was nevertheless efficient as its levels are indeed decreased in whole cell lysates. We confirm that Cisplatin reduces protein levels of chromatin-associated RNF113A.

6. Figure 3g. I do not understand why only one cell is positive for H₂AX considering that “Immunofluorescence analyses were conducted to define the subcellular localization of RNF113A in Cisplatin-treated A549 cells.” As the authors state in the figure legend.

Our answer:

We could see more than one cell positive for pH₂AX per field but the magnification of this image is quite high to better see each cell. About 15% of A549 cells are positive for pH₂AX when treated with Cisplatin for that time at that concentration (see quantification illustrated in Figure 2e). This fits with the image illustrated in Figure 3h).

7. In general, the IF images are quite poor. Specifically, I have concerns about the quantification of R-loops performed in Figures 4. Importantly, I am concerned because of the lack of nucleolar staining in control cells. This staining has been reported by several authors and is usually quite resistant to RNase H1 digestion (for expl. see Sollier, Mol Cell, 2014). I would suggest to more reliably quantify nucleoplasmic R-loop levels as described for ex. in Hodroji et al, EMBOJ 2017, i.e. using the DAPI signal to create a mask of the nucleus and then to determine the nuclear S9.6 signal intensity by subtracting the nucleolar signal (nucleolin staining) and analyzing the remaining S9.6 signal. Quantification should be performed for 100-200 cells per experiment.

Our answer:

All these experiments were re-done to meet Reviewer 2’s request (see our new Figures 4c and 5a). Quantifications were done according to Reviewer 2’s request.

8. P. 7 lines 147-148 “Accordingly, RNF113A deficiency enhanced the number of both

phospho-H2AX (pH2AX) and phospho-DNA-PKcs positive cells". As for R-loops quantification, the way the authors performed this experiment is puzzling. To quantify the total intensity in the whole nucleus then z-stacks should be analysed and not just one arbitrary plane. More appropriately, the authors should quantify the number of foci per cell in untreated and treated cells for at least 150 cells per experimental group.

Our answer:

All these experiments were also re-done to meet Reviewer 2's request (see our new Figures 2e and 2f). Quantifications were done according to Reviewer 2's request.

9. 148 (pDNA-PKcs) positive BZR-T33 cells, sAlso the way activation of the DDR was assessed by IF puzzleles me.

Our answer:

We do not precisely understand why this is puzzling.

10. All the experiments assessing splicing of specific transcripts lack quantifications (Figs. 5f, 6b, 6g, 7a).

Our answer:

Previous Fig. 5f has been removed from the manuscript because this analysis was distractive from our main message. All requested quantifications are now part of the revised Figures (see new Figures 7c, 7h and 8a).

11. The bioinformatics analysis performed on the RNAseq dataset is poor, in particular for what the splicing changes are concerned. How does RNF113A silencing affects splicing? Does it promote intron retention, inclusion/skipping of alternative exons?

Our answer:

To study the effect of RNF113A silencing on splicing, we have performed additional analyses with rMATS. This software is dedicated to the systematic analysis of the different forms of alternative splicing by looking at individual splicing events (skipped exons, mutually exclusive exons, retained intron, alternative 5' splicing site and alternative 3' splicing site). The results of this new analysis are reported in Figs. 6a-d and S7a-b of the revised manuscript. We have also clarified Methods and Results sections of the manuscript pertaining to the analyses of the RNAseq dataset. All these sections have been extensively rewritten. Altogether, we have performed three levels of analysis: (i) a differential expression at the gene level with DESeq2 evaluating three different models, (ii) a differential splicing analysis with rMATS focusing on individual splicing events of five categories, and (iii) a global intron-retention analysis at the gene level with dedicated homemade scripts. Our results

demonstrate that beyond an expected impact on the gene expression, the splicing is globally affected. Although some specific exon skipping events are significant upon RNF113A silencing, the main impact of the knockdown is a slight but significant decrease of the splicing efficiency.

12. *To further support the role of RNF113A on NOX1 activity the authors could complement .NOX1 in RNF113A deficient cells (Fig. 7d).*

Our answer:

We found it quite challenging to complement RNF113-depleted cells with anything as these cells quickly undergo cell death when further manipulated with transfecting reagents. Although this experiment would have indeed very interesting to do, we found it very difficult to do. We apologize about this.

13. *The methods section is incomplete. Several experimental approaches are not described. How was the phosphorylation of H2Ax quantified by IF? How were R-loops measured?*

Our answer:

The requested information is now provided in the revised version. For the quantification of R-loops, 10 fields per condition were analysed for quantification purposes and nuclei specks in a total of 50 nucleus in each experimental condition were counted.

Minor points

Sometimes the rational for specific experiments is not well explained, so that the paper is difficult to read for non-specialized readers. For example, it would not harm to explain the link between ERK1/2 and MEK1/2 (page 6, line 117-118).

Our answer:

Given the additional data that we show in the revised manuscript and because Reviewer 3 did not see the point of explaining that ERK1/2 activation controls RNF113A expression, we decided to remove these panels in the revised version. We agree that this observation is not essential for our publication. Therefore, it is not required to describe in more details the link between MEK1/2 and ERK1/2...

Spelling mistakes throughout: for ex. p19, line 435 the camptothecin concentrations are too high (mM or microM?) or p. 6, line 121: "RNF113A and C/EBP expression upon ERK1/2 inhibition (was) very highly similar,"

Our answer:

We corrected these spelling mistakes in the revised version. We thank Reviewer 2 for highlighting them.

Reviewer #3 (Remarks to the Author):

In the present manuscript the authors analyze the impact of RNF113A, a E3 ligase and a spliceosome subunit, in gene expression and DNA damage response (DDR) upon cisplatin treatment. The subject is of interest but data quality and presentation is questionable. In general terms the authors want to conclude that RNF113A is a splicing factor relevant for the DDR, but in my opinion they fail to properly conclude this although some of the data might be promising. Therefore, at least in its current form, the work is not suitable for publication in nature communications.

Major points:

- If RNF113A controls the pool of NHEJ factors recruited to damaged DNA, then the authors should work with cells arrested in G1, so as to avoid replication issues, and use a true DSB inductor, not cisplatin, to be able to say that RNF113A controls the pool of the recruited NHEJ factors. This applies also to other relevant results such as the proposed differential binding of RNF113A to mRNAs after damage (figure 4) or the effect on cell survival upon knock down / overexpression of RNF113A (figure 2).

Our answer:

The main conclusion of our paper is that RNF113A acts as a spliceosome subunit to promote cell survival upon DNA damage. We indeed show that the recruitment of NHEJ factors to damaged DNA is deregulated upon RNF113A deficiency but we cannot rule out at this point that the mechanisms are indirect, i.e. due to defective splicing of multiple pro-survival candidates. Note that we focused on Cisplatin as this drug nicely induces RNF113A expression and this is the initial reason why we concentrated on it. Cisplatin has also been largely used to treat patients suffering from lung cancer, the pathological context in which we address the biological roles of RNF113A. In order to meet Reviewer 3's wish, we also treated control and RNF113A-depleted A549 cells with Etoposide, a true DSB inducer and we now show that RNF113A also protects from Etoposide-dependent cell death (see our new Figure S3).

- When there is no over-expression is not clear at all that cisplatin reduces the amount of RNF113A to oligodT bounded RNAs (Fig 4a, lines 1 and 3) since it looks like there is more binding (lines 1 vs 3) but also the input is affected (lines 5 vs 7). Conclusions about differential binding in normal or cisplatin treated cells should be done in cells with no RNF113A overexpression. Also, proper quantitation of nuclear and cytoplasmic RNF113A in control and cisplatin treated no over-expressing cells should be done.

Our answer:

We carried out the requested experiment using parental cells (see Figure 4a and 4b). Our conclusions remain valid. The requested quantification was done in the graph illustrated below. We set the density of signal in cytoplasmic fraction of non-treated cells to 100%.

- *Focusing on splicing and alternative splicing, the authors should just present the percentage of alternative exon cassette inclusion (and maybe another forms of AS that are less common) and/or unspliced transcripts upon cisplatin and RNF113A knock down in the whole transcriptome. Then they should specifically analyze the subset of relevant genes i.e. death related genes. The way the authors present the data is rather confusing and this is not just in figure 5, but throughout the hole text (see below).*

Our answer:

To study the effect of RNF113A silencing on splicing, we have performed additional analyses with rMATS. This software is dedicated to the systematic analysis of the different forms of alternative splicing by looking at individual splicing events (skipped exons, mutually exclusive exons, retained intron, alternative 5' splicing site and alternative 3' splicing site). The results of this new analysis are reported in Figures 6a-d and S7a-b of the revised manuscript. We have also clarified both Methods and Results sections of the manuscript pertaining to the analyses of the RNAseq dataset. All these sections have been extensively rewritten. Altogether, we have performed three levels of analysis: (i) a differential expression at the gene level with DESeq2 evaluating three different models, (ii) a differential splicing analysis with rMATS focusing on individual splicing events of five categories, and (iii) a global intron-retention analysis at the gene level with dedicated homemade scripts. Our results demonstrate that beyond an expected impact on the gene expression, the splicing is globally affected. Although some specific exon skipping events are significant upon RNF113A silencing, the main impact of the knockdown is a slight but significant decrease of the splicing efficiency. Given the global nature of this impact on intron retention, death related genes are affected comparably to other types of genes. This was tested in a Gene Set Enrichment Analysis that did not show any enrichment.

- *The authors say that RNF113A is more expressed upon cisplatin but also more localized in the cytoplasmic fraction (while splicing takes place in the nucleus). They also say that upon cisplatin, RNF113A is less bound to pre-mRNAs but, at the same time, RNF113A is necessary to properly express key mRNAs isoforms like, for instance, in the Nupr1 gene. To this referee this is not clear at all.*

Our answer:

We indeed confirm that the pool of cytoplasmic RNF113A increases upon Cisplatin stimulation while its nuclear pool decreases (see our experiment illustrated in Figure 3c as well as our new Figure 5d). This observation is also relevant in BEAS-2B cells, which are derived from a normal bronchial epithelium (see our new Figure S4b). We now show that both SF3B1 and SF3B2 disengage from the chromatin upon DNA damage (see our new Figure S4b). This observation is not in contradiction with the fact these proteins promote RNA splicing in the nucleus as we show that these splicing factors can still be found in the nucleus, even after 24 hours of stimulation with Cisplatin. We have discussed this issue in the revised discussion (see the text highlighted in yellow).

Minor points:

The manuscript have many errors and omissions that makes data interpretation difficult. Bellow some examples,

Fig 1 a, what is the ratio RNF113A / actin?

Our answer:

We carried out densitometry analyses from our western blots for all experimental conditions. The ratio RNF113/ β -Actin was subsequently established using these densitometry values.

Fig 1b, what is the purpose of this figure? What is the localization of RNF113A in normal cells? Is there any function of RNF113A in cytoplasm?

Our answer:

We carried out some western blot analyses with both nuclear and cytoplasmic extracts from BEAS-2B cells, which are derived from a normal bronchial epithelium and we now show that the pool of cytoplasmic RNF113A increases upon DNA damage while the pool of nuclear RNF113A decreases, as previously showed in A549 cells (Figure S4b). Therefore, RNF113A can be found both in the nucleus and in the cytoplasm and its shuttling from the nucleus to the cytoplasm upon DNA damage is observed in multiple experimental models. So far, we cannot define any established role of RNF113A in the cytoplasm. We can only state that RNF113A may move into the cytoplasm as a splicing factor bound to RNAs upon DNA damage.

Fig 1 d/e, now hsp90 is the loading control but in fig 1a was actin. The authors should use the same factor as a loading control throughout the manuscript. Moreover, overexposed western blots should be replaced.

Our answer:

We decided to use HSP90 as a loading control throughout the manuscript but could not do it for the western blot using extracts from clinical cases of lung cancer showed in Figure 1a as HSP90 is actually overexpressed in a variety of cancer types (see for example these publications: Fuller KJ et al. Cancer and the heat shock response. European Journal of Cancer. 1994; 30:1884-1891; Biaoxue R et al. Upregulation of Hsp90-beta and annexin A1 collerates with poor survival and lymphatic metastasis in lung cancer patients. Journal of Experimental & Clinical Cancer Research. 2012; 31, 70). It would have been a lot of work to redo all our western blots using β -Actin as a loading control. We think that this strategy does not negatively impact on the conclusions that we raised in this study. Please find hereafter our anti-HSP90 western blot carried out with protein extracts from normal and cancer tissues.

Regarding the overexposed blots, most of them were anti-ERK1/2 panels which were removed in the revised version in order to meet your request (see below). We now provide a shorter exposure time for the anti-C/EBP β western blot illustrated in former Figure 1k (now Figure 1l).

Fig 1 d/e, cisplatin treatment enhanced RNF113A in cancer cells lines, and in non-transformed cells? Is there any point in stressing “in cancer cells” or it’s a general phenomenon after cisplatin treatment? What about Ku80 wb? Is not commented anywhere in the text.

Our answer:

We initially wanted to explore whether Ku80 expression was changed or not upon RNF113A deficiency but we agree that it is not essential for the message of this figure so we removed these panels in the revised version. RNF113A expression is indeed induced by Cisplatin in normal human dermal fibroblasts as well (see our new Figure S1d). We added these comments in the text.

Fig 1 i/j Unless there is a point in analyzing ERK1/2 dependency in RNF113A expression in different cells lines the figure should be removed.

Our answer:

We removed the panels on the link between ERK1/2 and RNF113A expression in the revised version. We agree that it is not essential in our study.

Fig 2a, the legend states that the western was carried out 20 hours after cisplatin treatment but the figure shows a time course.

Our answer:

We apologize for this mistake which was corrected in the revised version.

Fig 2b, only one experiment should be shown, otherwise the figure is misleading.

Our answer:

Following Reviewer 3's suggestion, we only show FACS data from one representative experiment in the revised figure 2.

Fig 2d, the authors say that "RNF113A deficiency increases the inhibitory effect of Cisplatin on the clonogenicity of A549 cells". Is there any interaction between cisplatin and shRNA treatment? Can the authors really say that the cisplatin effect is increased upon RNF113 deficiency? It is clear that there are less cells in shRNA treated cells with no cisplatin.

Our answer:

Given the fact that our revised paper includes numerous experiments and because this clonogenic assay does not add any additional molecular insights into our study, we decided to remove this experiment in the revised version in order to exclusively concentrate on major findings.

Fig 3d, In the chromatin fraction the RNF113A levels are wrong or the legend is wrong.

Our answer:

We were indeed surprised to notice that RNF113A deficiency did not have much impact on protein levels of RNF113A associated to the chromatin. Our depletion of RNF113A was nevertheless efficient as its levels are indeed decreased in whole cell lysates.

Fig 4a/b, RNaseA is written in different ways in figures 4a and 4b.

Our answer:

This mistake has been corrected in the revised version.

Fig 5b, Figure legends and text doesn't match. Control bars are #2 and #4???? Or is the other way around?

Our answer:

The legend has been corrected in the new figure.

Fig 5f, this is wrong, there is no alternative intron (as stated in the text, page 11 line 254) nor alternative exon (as written in the figure) but alternative promoters as is always the case when exon 1 is different between two transcripts sharing the rest of the mRNA sequences. The end point PCR showed at the bottom right of the figure shows no quantitation and therefore is not clear if RNAseq results are validated or not. Of course, the authors should use a qPCR approach to be able to talk about different quantities of RNF116 transcript upon shRNA treatment or cisplatin treatment. Moreover, and once again, the legend of the figure is wrong since it is shown “cisplatin cisplatin” on top of the gel where the correct sentence is probably “control cisplatin”.

Our answer:

As new analyses of alternative splicing events have been added to figure 6, the former panel 5f has been removed.

Fig 6b, once again the figure is hard to understand. It is not clear what is the treatment of each sample with respect to untreated or cisplatin treated and shRNAs. A ratio largo/short isoforms should be presented.

Our answer:

Figure 7b (former Figure 6b) now presents the results of a new analysis with rMATS assessing notably SAT1 exon 4 inclusion levels.

Fig 6f, why the authors are show 8 bars in the graph? What is the difference between the first and the last 4 bars? I see no explanation in the text or in the figure legend. This is happening quite often in this work, understanding figures is not an easy task.

Our answer:

Figure 7f (former Figure 6f) has been replaced by a panel presenting NUPR1 mRNA quantification at the gene level based on DESeq2 analysis.

Fig 8a, there is no loading control? The how can the authors and readers properly conclude that MCL is degraded in the absence of RNF113A?

Our answer:

We now provide an anti-HSP90 western blot for normalization purposes in Figure 9a (former Figure 8a). We indeed demonstrate that MCL-1 is destabilized upon RNF113A deficiency. Indeed, MCL-1 half-life is increased upon RNF113A overexpression (Figure 9c). Likewise, the polyubiquitination of MCL-1 is enhanced upon RNF113A deficiency (Figure 9e). As MG132 restores MCL-1 protein levels in RNF113A-depleted cells (Figure 9d), our data suggest

that RNF113A promotes MCL-1 expression, at least by limiting its polyubiquitination and subsequent proteasome-dependent degradation.

Reviewers' comments:

Reviewer #1 (Remarks to the Author):

The revised manuscript has partially addressed the previous criticisms. However, it is still very confusing in the organization and some issues and statements appear not supported by the data.

Specific comments:

Page 8, line 167: "Having established a role of RNF113A in DNA repair". As already mentioned in the previous revision, the role of RNF113A in DNA repair was already established in a previous Nature paper. Here they authors confirm that observation and extend it to lung cancer cells. The sentence should be removed or rephrased.

Figure 4b: unless this panel was mislabeled, lane 2 in which cisplatin was added shows stronger recruitment of both RNF113A and SF3B1 compared to lane 4 of untreated cells, not impaired binding as described in the text. Please clarify.

Figure 4c-f: to state that RNF113A is required for assembly of spliceosomal components and to prevent R-loops formation, the authors should rule out that these events are not the consequence of the increased apoptosis that they observed upon depletion of RNF113A. If they induce apoptosis by a means that does not modulate RNF113A (for instance etoposide treatment), do they observe increase in R-loops and disassembly of the spliceosome? Indeed, it is known that ATM also induces detachment of the spliceosome from the chromatin (Tresini et al., Nature 2015) and that splicing is impaired in apoptotic cells.

Page 11: "We noticed that RNF113A depleted cells accumulated RAD51 pre-mRNAs and showed less mature RAD51 mature mRNAs, especially after Cisplatin stimulation (Fig. S6a). Therefore, RNF113A is required for the proper splicing of RAD51". The data shown in Fig. S6a do not support this conclusion and if they want to state that RNF113A is required for proper splicing of RAD51 they should perform a more accurate analysis of the splicing of this gene. In light of the genome-wide analysis shown in Fig. 6, this piece of information can be deleted.

Fig. 6: the representation of the splicing results are not very straightforward to understand. A simpler and clearer representation of the splicing changes in the different conditions would help. What is the overlap between intron-retaining genes and downregulated genes? Are these transcripts targeted for degradation?

Fig. 9 and Fig. 10: after the description of the role of RNF113A in splicing regulation, the authors show that this protein also regulates the stability of MCL-1 protein. What is the link between the two stories? As presented (already indicated in the first revision), the work appears as a collection of information on RNF113A, not always and necessarily linked.

Reviewer #2 (Remarks to the Author):

The authors addressed most of my concerns and I find the analysis of the RNAseq dataset very careful and interesting. However, because of the new results the manuscript is now quite unbalanced. I would suggest to reorganize the Introduction, which deals mostly with DNA damage and the DDR, and the Discussion to better link the results concerning the characterization of the role of RNF133A in the DDR with the splicing section.

In addition, although the effects of RNF113A depletion on the DDR may be indirect I still believe that a characterization of the recruitment of the NHEJ factors that are affected by RNF133A silencing at DSBs by CHIP in DIVA cells could be informative, since it is equally possible that upon

DNA damage RNF113A may act on a subset of DDR factors or on histones, as other Ring Finger ubiquitin ligases. Unfortunately, the authors only monitored H2AX which only confirmed an increase of damage upon RNF113A silencing, similar to IF results. Therefore the conclusion "RNF113A controls the pool of NHEJ factors recruited to damaged DNA " is still not supported (page 9, line 188).

Reviewer #3 (Remarks to the Author):

The authors did a great job and the majority of my concerns have been addressed. I think that the work is suitable for publication.

Reviewer #1 (Remarks to the Author):

The revised manuscript has partially addressed the previous criticisms. However, it is still very confusing in the organization and some issues and statements appear not supported by the data.

Our answer:

Reviewer 1 should have specifically mentioned which statement was not supported by our data as we do not agree with this statement. All our initial conclusions remain valid and we believe that we describe multiple experiments to demonstrate that RNF113A promotes DNA repair as a spliceosome subunit, which is totally new and has never been reported in any publication, including the Nature paper mentioned by Reviewer 1. More precise answers to his/her comments can be found here after.

Specific comments:

Page 8, line 167: “Having established a role of RNF113A in DNA repair”. As already mentioned in the previous revision, the role of RNF113A in DNA repair was already established in a previous Nature paper. Here they authors confirm that observation and extend it to lung cancer cells. The sentence should be removed or rephrased.

Our answer:

We had already seen that RNF113A was promoting DNA repair at the time the Nature paper came out. As we independently made this observation, which is the starting point for all mechanistical studies that we now report in our manuscript, we felt that it made sense to write this statement. This is perfectly fine as we mention the Nature paper in our manuscript of course.

Figure 4b: unless this panel was mislabeled, lane 2 in which cisplatin was added shows stronger recruitment of both RNF113A and SF3B1 compared to lane 4 of untreated cells, not impaired binding as described in the text. Please clarify.

Our answer:

Our statement is correct. Unfortunately, in the process of generating the last version of this Figure, both « + » were misplaced. Data with treated cells are illustrated in both lanes 3 and 4. We now provide the corrected version.

Figure 4c-f: to state that RNF113A is required for assembly of spliceosomal components and to prevent R-loops formation, the authors should rule out that these events are not the consequence of the increased apoptosis that they observed upon depletion of RNF113A. If they induce apoptosis by a means that does not modulate RNF113A (for instance etoposide treatment), do they observe increase in R-loops and disassembly of the spliceosome? Indeed, it is known that ATM also induces detachment of the spliceosome from the chromatin (Tresini et al., Nature 2015) and that splicing is impaired in apoptotic cells.

Our answer:

All our data showing R-Loops are exclusively done in control versus RNF113A-depleted cells that were not treated with any DNA damaging agent. Therefore, there is not need to use Etoposide as a drug that does not induced RNF113A expression as we already know that R-loops will increase upon RNF113A deficiency. Regarding the issue between the loss of RNF113A and cell apoptosis, we experimentally addressed this issue by treated our cells with ZVAD, a caspase inhibitor, and by looking at consequences on splicing events upon DNA damage. As expected, ZVAD prevented Caspase 3 activation upon Cisplatin treatment in RNF113A-depleted cells (see our **new Figure 6c**, compare lanes 6 and 8). Interestingly, the splicing defects seen upon Cisplatin treatment in RNF113A-depleted cells was still observed in cells treated with ZVAD (Fig. 6c, lower panels). This conclusion also applied to SF3B2-depleted cells which also showed some splicing deregulations of SAT1. Here again, ZVAD did not prevent these splicing deregulations upon DNA damage (see our **new Figure 6d**). Similar experiments were conducted on other candidates such as Rad51 and RNF8. We also pretreated A549 cells with ZVAD and assessed Rad51 and RNF8 splicing upon Cisplatin stimulation in both control and RNF113A-depleted cells. ZVAD did not impact on the accumulation of both Rad51 and RNF8 pre-mRNAs upon RNF113A deficiency in Cisplatin-treated cells (see our **new Figures S6a and S6b**, respectively). Therefore, apoptosis is the consequence rather than the cause of the splicing deregulations seen upon RNF113A deficiency in cells subjected to a DNA damaging agent.

Page 11: “We noticed that RNF113A depleted cells accumulated RAD51 pre-mRNAs and showed less mature RAD51 mature mRNAs, especially after Cisplatin stimulation (Fig. S6a). Therefore, RNF113A is required for the proper splicing of RAD51”. The data shown in Fig. S6a do not support this conclusion and if they want to state that RNF113A is required for proper splicing of RAD51 they should perform a more accurate analysis of the splicing of this gene. In light of the genome-wide analysis shown in Fig. 6, this piece of information can be deleted.

Our answer:

We took the same primers as the ones described in a publication (now reference 40) to first make the point that RNF113A deficiency was linked to a splicing defect of a given candidate, namely Rad51. This initial results prompted us to carry out a genome-wide analysis now illustrated in Figure 5. This data on Rad51 splicing is now illustrated as a supplementary figure panel (Figure S6a) in which we also assessed the effect of the Caspase inhibitor ZVAD on Rad51 splicing.

Fig. 6: the representation of the splicing results are not very straightforward to understand. A simpler and clearer representation of the splicing changes in the different conditions would help. What is the overlap between intron-retaining genes and downregulated genes? Are these transcripts targeted for degradation?

Our answer:

We carried out several detailed bioinformatical analyses to make the point that thousands of splicing events were deregulated upon RNF113A deficiency. We strongly believe that our panels nicely illustrate the complexity of what we see. There is not other ways to illustrate such complexity than by showing plots and histograms as we did. Regarding the overlap between intron-retaining genes and downregulated genes, downregulated genes could be a indirect consequence of differentially spliced candidates that regulate gene transcription. This specific issue was not raised by reviewer 1 in the first round of reviewing and is beyond the scoop of this manuscript. For sure, all the transcripts whose splicing is defective upon RNF113A are not subjected to degradation as some protein proteins are still produced from these transcripts in RNF113A-depleted cells (see our Figures 6^e and 8c).

Fig. 9 and Fig. 10: after the description of the role of RNF113A in splicing regulation, the authors show that this protein also regulates the stability of MCL-1 protein. What is the link between the two stories? As presented (already indicated in the first revision), the work appears as a collection of information on RNF113A, not always and necessarily linked.

Our answer:

We specifically highlighted the fact that RNF113 was promoting MCL-1 stability as a spliceosome subunit. Indeed, we show in Figure 9h that SF3B2 deficiency also decreases MCL-1 protein levels. This is a strong point to conclude that RNF113A regulates MCL-1 stability as a spliceosome subunit. Importantly, we also show in Figure 10c that levels of pro-survival versus pro-cell death MCL-1 transcripts are changing upon RNF113A deficiency, which indicates that MCL-1 is a direct target of RNF113A in Cisplatin-resistant cells.

We further looked at multiple potential mechanisms by which RNF113A promotes MCL-1 stability in this revised version. We first ruled out the possibility that RNF113A directly targets MCL-1 polyubiquitination in a non degradative manner in parental cells (data not shown). We next looked at expression levels of all candidates known to regulate the degradative polyubiquitination of MCL-1 and found that USP9X was actually downregulated in both RNF113A and SF3B2-depleted cells (see our **new Figures 9f and 9h**, respectively). This new observation may contribute to the destabilization of MCL-1 seen upon RNF113A or SF3B2 deficiency. We still discuss about other potential molecular mechanisms (ROS production and MULE1 activity) in the revised discussion.

Regarding Reviewer 1's statement on the « collection of information on RNF113A », we actually believe that our data on RNF113A, SF3B2 and MCL-1 support the notion that RNF113A is acting as a spliceosome subunit. To demonstrate this statement even more convincingly in this revised version, we first showed that SF3B2-depleted cells accumulated more Rad51 pre-mRNAs after Cisplatin treatment, similarly to RNF113A-depleted cells (see our **new Figure S6a**). These SF3B2-depleted cells also showed some splicing deregulations of SAT1 (see our **new Figure 6d**). We also showed that Noxa1 ΔSH3 protein levels were more elevated upon Cisplatin stimulation in both RNF113A- and SF3B2-depleted cells (see our Figure 8c). Therefore, our data indicate that RNF113A or SF3B2 deficiency has very similar consequences on splicing deregulations seen upon DNA damage in lung cancer cells. Again, these multiple experiments make the point that RNF113A is a spliceosome subunit and this explains the consequences that we see on MCL-1 protein levels.

Reviewer #2 (Remarks to the Author):

The authors addressed most of my concerns and I find the analysis of the RNAseq dataset very careful and interesting. However, because of the new results the manuscript is now quite unbalanced. I would suggest to reorganize the Introduction, which deals mostly with DNA damage and the DDR, and the Discussion to better link the results concerning the characterization of the role of RNF133A in the DDR with the splicing section. In addition, although the effects of RNF113A depletion on the DDR may be indirect I still believe that a characterization of the recruitment of the NHEJ factors that are affected by RNF133A silencing at DSBs by ChIP in DIVA cells could be informative, since it is equally possible that upon DNA damage RNF113A may act on a subset of DDR factors or on histones, as other Ring Finger ubiquitin ligases. Unfortunately, the authors only monitored H2AX which only confirmed an increase of damage upon RNF113A silencing, similar to IF results. Therefore the conclusion “RNF113A controls the pool of NHEJ factors recruited to damaged DNA “ is still not supported (page 9, line 188).

Our answer:

We updated our discussion by adding the following paragraph : « Moreover the co-localization of RNF113A with DNA-PKcs in Cisplatin-treated A549 cells as well as a defective engagement of pH₂AX to DSB sites in depleted DIVA cells suggests its recruitment at the extremities DNA double-strand breaks where it could act as an E3 ligase to directly promote DNA repair. Alternatively, RNF113A may indirectly promote this process as a spliceosome subunit. This hypothesis is experimentally supported by our demonstration that RNF8, which promotes histone polyubiquitination and the recruitment of 53BP1 and BRCA1 repair proteins to double strand breaks⁵⁴, is a target of RNF113A as a spliceosome subunit. Therefore, RNF113A appears to be involved in DNA repair through both direct and indirect mechanisms”.

We agree with Reviewer 2 that most if not all our data define RNF113A as a spliceosome subunit which promotes cell survival. Indeed, RNF113A may indirectly regulate the DDR. We now provide a revised version in which we increased our data describing RNF113A as a spliceosome subunit (cf our **new Figures 6 and S6**). We also updated the Introduction by giving more emphasis on the spliceosome (cf our **first paragraph**). We also removed some words on the DDR in the Introduction. Note that we still kept some words on the DDR in the introduction as our manuscript still includes some data on DNA-PKcs phosphorylation and on the recruitment of DDR actors on DNA breaks in both control and RNF113A-depleted cells.

Regarding the issue with the recruitment of NHEJ factors, we favor the hypothesis that effects of RNF113A on DDR are indirect. We could investigate whether or not RNF113A regulates the recruitment of NHEJ factors but these experiments would not tell us whether RNF113A directly acts in the nucleus in these processes. Let's also note that our data illustrated in Figure 3d already tell us that RNF113A-depleted cells show more Ku70 and

Ku80 in chromatin fractions upon Cisplatin treatment. We actually carried out several new experiments in DIVA cells to meet one concern raised by Reviewer 2 in the previous round of reviewing but we rather believe that the novelty of our paper is in the demonstration that RNF113A acts as a spliceosome subunit.

Reviewer #3 (Remarks to the Author):

The authors did a great job and the majority of my concerns have been addressed. I think that the work is suitable for publication.

Our answer:

We thank Reviewer 3 for his/her constructive comments during the reviewing process of our manuscript.

REVIEWERS' COMMENTS:

Reviewer #1 (Remarks to the Author):

The authors have substantially addressed the criticisms raised to the previous version of the manuscript, which now appears improved.

Reviewer #2 (Remarks to the Author):

Most of the suggestions have been addressed in this revised version thus I have no further queries. I think that the manuscript now fulfils the requirements for a publication.